# Lifelong Learning with Behavior Consolidation for Vehicle Routing

**Jiyuan Pei[1], Yi Mei[1], Jialin Liu[2], Mengjie Zhang[1], Xin Yao[2]**

[1]Center of Data Science and Artificial Intelligence & School of Engineering and
    Computer Science, Victoria University of Wellington, Wellington, New Zealand

[2]School of Data Science, Lingnan University, Hong Kong SAR, China

`jiyuan.pei@vuw.ac.nz`, `{yi.mei,mengjie.zhang}@ecs.vuw.ac.nz`,
`{jialin.liu,xinyao}@ln.edu.hk`

## Abstract

Recent neural solvers have demonstrated promising performance in learning to solve routing problems. However, existing studies are primarily based on one-off training on one or a set of predefined problem distributions and scales, i.e., tasks. When a new task arises, they typically rely on either zero-shot generalization, which may be poor due to the discrepancies between the new task and the training task(s), or fine-tuning the pretrained solver on the new task, which possibly leads to catastrophic forgetting of knowledge acquired from previous tasks. This paper explores a novel lifelong learning paradigm for neural VRP solvers, where multiple tasks with diverse distributions and scales arise sequentially over time. Solvers are required to effectively and efficiently learn to solve new tasks while maintaining their performance on previously learned tasks. Consequently, a novel framework called **L**ifelong **L**earning **R**outer with **B**ehavior **C**onsolidation (LLR-BC) is proposed. LLR-BC consolidates prior knowledge effectively by aligning behaviors of the solver trained on a new task with the buffered ones in a decision-seeking way. To encourage more focus on crucial experiences, LLR-BC assigns greater consolidated weights to decisions with lower confidence. Extensive experiments on capacitated vehicle routing problems and traveling salesman problems demonstrate LLR-BC's effectiveness in training high-performance neural solvers in a lifelong learning setting, addressing the catastrophic forgetting issue, maintaining their plasticity, and improving zero-shot generalization ability.

## 1 Introduction

Neural solvers have gained significant attention for their remarkable performance on vehicle routing problems (VRPs) (Vinyals et al., 2015; Bello et al., 2017; Bengio et al., 2021; Bogyrbayeva et al., 2024). By training on one or multiple specific problem distributions and scales, i.e., tasks, simultaneously, neural solvers can learn to construct high-quality solutions or improve existing solutions efficiently for problems drawn from the same distribution and of the same scale as the training tasks (Kool et al., 2019; Kwon et al., 2020; Lu et al., 2020; Tang & Yao, 2024). However, in real-world scenarios, unpredictable problems with new distributions or scales often arise

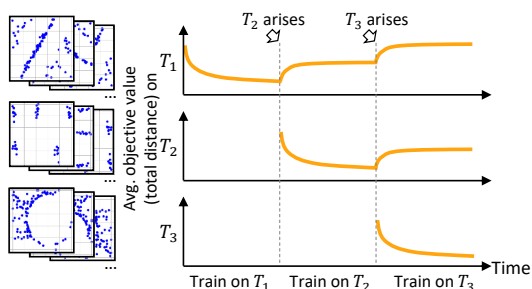

Figure 1: Conceptual demonstration of catastrophic forgetting while fine-tuning on sequential, new tasks with different distributions and scales.

over time. For example, a daily delivery service provider receives different numbers of orders with new patterns of delivery locations and demands due to unpredictable consumer behavior, business expansion, and new shopping trends. In such scenarios, it is challenging to develop a universal solver that can perform effectively across all possible problem distributions and scales after one-off

training (Liu et al., 2023). Some recent studies (Jiang et al., 2022; Bi et al., 2022; Gao et al., 2024; Wang et al., 2024b; Xiao et al., 2025) address unpredictable new problems by enhancing the generalization ability of trained solvers. However, the generalization would have a boundary, and on new problems that differ substantially from the training ones, these methods still require additional learning to enhance problem-solving performance (Zhou et al., 2023). A common approach is to fine-tune the trained solver on a new task (Bengio et al., 2021; Zhou et al., 2023); nonetheless, it leads to catastrophic forgetting of knowledge acquired from previous tasks (Khetarpal et al., 2022). Solver's performance on problem instances of the previous tasks will decrease significantly (cf. Figure 1), as the model parameters will be overwritten during training on a new task (Wang et al., 2024c). These issues highlight the need for neural solvers capable of lifelong learning, enabling them to continuously acquire knowledge from unforeseen tasks while retaining generic knowledge and their performance on previous tasks.

Existing studies on lifelong learning for neural VRP solvers (Li et al., 2024; Feng et al., 2025) are largely confined to highly specific scenario settings: tasks solely differing in scale or distance metric, the task order is fixed and known, and the generation of new problem instances is controllable. Furthermore, they are mainly dedicated to the solver's performance at the end of the lifelong learning process after being trained on all training tasks. As lifelong learning is an ongoing process (possibly without any end), the solver's performance at each time point of the lifelong learning process is also important. Their proposed methods rely on actively generating and training on new problem instances from previous tasks, making them inapplicable when the generation of new instances is uncontrollable, as is the case studied in this work.

In contrast, this work targets more realistic and general lifelong learning scenarios, simultaneously accommodating changes in both scale and distribution across tasks, without assuming any prior knowledge of the task order or any control over problem instance generation. We propose a novel method for lifelong learning of neural VRP solvers, called **L**ifelong **L**earning **R**outer with **B**ehavior **C**onsolidation (LLR-BC). LLR-BC focuses on addressing the catastrophic forgetting issue that arises during training across new tasks arising sequentially, which is commonly overlooked in prior neural VRP solver research. It buffers and revisits experiences from previous tasks to retain acquired knowledge. To efficiently utilize the limited experience memory, LLR-BC weights experiences based on decision confidence, emphasizing more crucial ones. Given that small changes in action probability distribution, especially under low confidence, can alter decisions and drastically affect the constructed route, LLR-BC minimizes reverse Kullback–Leibler divergence to consolidate behavior in a decision-seeking way. Extensive experiments on capacitated vehicle routing problems (CVRPs) and traveling salesman problems (TSPs), equipped with a comprehensive metrics set to evaluate the solver during the ongoing lifelong learning process, under various task sequences and base neural solvers show that LLR-BC effectively mitigates forgetting, maintains plasticity, and improves zero-shot generalization by accumulating transferable knowledge over time.

Our major contributions include: i) we introduce LLR-BC, a general lifelong learning framework enabling neural VRP solvers to learn from different tasks sequentially and address catastrophic forgetting effectively; ii) we propose Confidence-aware Experience Weighting (CaEW) to prioritize crucial experiences, as a module of LLR-BC for enhancing the effectiveness of experiences utilization; iii) we propose Decision-seeking Behavior Consolidation (DsBC), which works in LLR-BC to preserve past behaviors by minimizing the discrepancy between buffered and current solver behaviors on stored states, with an emphasis on replicating past decisions; iv) the effectiveness of LLR-BC in is validated through extensive experiments, supported by detailed analysis and discussion.

## 2 RELATED WORK

### 2.1 NEURAL VRP SOLVERS

Machine learning, particularly deep reinforcement learning (DRL), enables neural solvers to learn VRP-solving strategies directly from problem-solving experience, eliminating the need for hand-crafted heuristics (Bengio et al., 2021). Existing methods are categorized as neural construction or neural improvement approaches (Bi et al., 2022; Kong et al., 2024). Neural construction methods build solutions from scratch by sequentially selecting the next node in an autoregressive fashion. Notable examples such as POMO (Kwon et al., 2020) and DualOpt (Zhou et al., 2025) generate high-quality solutions within seconds, rivaling strong heuristics like LKH3 (Helsgaun, 2017). In

contrast, neural improvement methods learn to enhance existing solutions, either by configuring (Wu et al., 2022; Ma et al., 2023) or selecting (Lu et al., 2020; Pei et al., 2025b; Chen et al., 2025; Guo et al., 2025) among predefined improvement heuristics. Although improvement methods have the potential to further improve solution quality by increasing the number of iterations, this also leads to higher computational cost. This paper focuses on lifelong learning for neural construction methods, due to their favorable balance between effectiveness and efficiency.

## 2.2 CROSS-DISTRIBUTION CROSS-SCALE GENERALIZATION

Early studies of neural VRP solvers typically focus on solving a single task, i.e., problems with fixed scale and one given distribution (e.g., uniform) (Kool et al., 2019; Kwon et al., 2020), resulting in poor generalization across distributions and scales (Bi et al., 2022; Xiao et al., 2025), which is often encountered in real-world applications. To address this issue, recent studies focus on improving cross-distribution or cross-scale performance. DROP (Jiang et al., 2022) improves cross-distribution performance by training across multiple distributions and optimizing for worst-case task performance. AMDKD (Bi et al., 2022) distills knowledge from multiple task-specific teacher models into a student model, achieving improvement in zero-shot generalization. ELG (Gao et al., 2024) separates local and global policies and, despite training only on uniform problems of scale 100, generalizes well across other tasks. Xiao et al. (2025) proposes two network modules to handle different scales and distributions, respectively, improving generalization. INViT (Fang et al., 2024) involves multiple encoders and decoders with multiple views to improve generalization. Omni (Zhou et al., 2023) applies meta-learning to learn a solver that can quickly adapt to a new task.

However, existing studies largely rely on one-off training on a predefined task set. New tasks are typically handled via zero-shot generalization or independent fine-tuning, without leveraging knowledge accumulated across new tasks, and with limited consideration of catastrophic forgetting. Consequently, their effectiveness in sequential task learning remains limited. In realistic settings, tasks arrive continuously and previously learned tasks can inform future adaptation, motivating the development of lifelong neural VRP solvers that incrementally learn from a stream of diverse tasks.

## 2.3 LIFELONG LEARNING

Lifelong learning, also referred to as continual learning (Wang et al., 2024c), involves training a model to sequentially learn multiple tasks, progressively improving its capabilities in a manner analogous to human learning throughout life (Thrun, 1998; Khetarpal et al., 2022). Each task is characterized by a distinct data distribution (Wang et al., 2024c). A key challenge lies in balancing *plasticity*, i.e., the ability to acquire knowledge of new tasks, with *stability*, i.e., the capacity of retaining knowledge from past tasks (Parisi et al., 2019). Insufficient stability leads to *catastrophic forgetting*, where the performance on earlier tasks degrades after training on new tasks.

Several strategies have been proposed to mitigate catastrophic forgetting. Training task-specific model components is an example (Wu et al., 2021; Ebrahimi et al., 2020), but this increases storage costs and poses challenges in selecting the correct model when the task of the test data is unknown. Regularization of parameter updating to preserve prior knowledge is also widely used (Kirkpatrick et al., 2017; Zenke et al., 2017). However, this can hinder learning on new tasks. A further widely adopted method is experience replay, where past experiences on previously learned tasks are stored and revisited during training on a new task (Isele & Cosgun, 2018; Buzzega et al., 2020). Nonetheless, identifying and retaining the most crucial experiences to improve memory efficiency and minimize interference with learning new tasks often requires problem-specific designs.

To the best of our knowledge, only two existing works have studied lifelong learning for neural VRP solver (Li et al., 2024; Feng et al., 2025). However, their studies focused on rather restricted scenarios: (i) tasks differ only in scale (Li et al., 2024; Feng et al., 2025) or distance metric (Feng et al., 2025), (ii) task order is fixed and known under strong assumptions (e.g., task scale gradually increases), and (iii) problem instance generation is controllable so that new instances of previously learned tasks can always be actively generated, making these methods inapplicable to scenarios with uncontrollable instance streams. Studies of more generic and practical scenarios are desired.

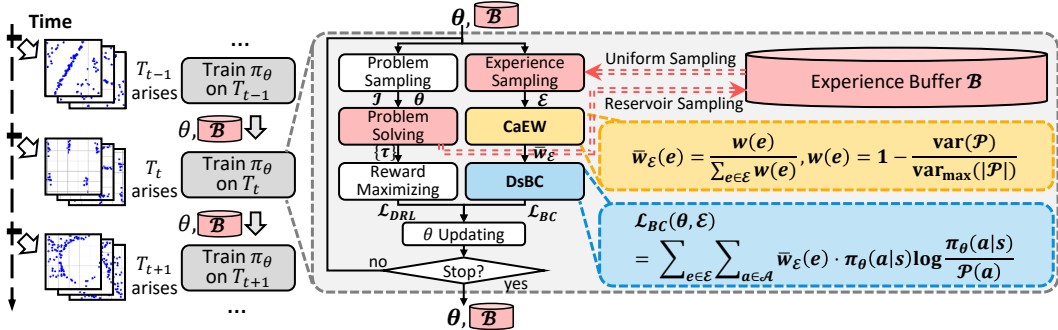

Figure 2: LLR-BC in the lifelong learning scenario where new tasks with different distributions and scales sequentially arise over time. $T_t$: the task at time $t$. $\pi_\theta$: solver with model parameters $\theta$. $\mathcal{B}$: experience buffer. $\mathcal{J}$: problem instances. $\{\tau\}$: problem solving trajectories. $\mathcal{E}$: experiences. $w_\mathcal{E}$: weights of experiences. $a$: action, i.e., node to visit. $s$: state. var: variance. $\mathcal{P}$: behavior.

## 3 LIFELONG LEARNING ROUTER WITH BEHAVIOR CONSOLIDATION

We consider the scenarios where problem instance generation is uncontrollable and only new instances of the current task are generated for training, while future tasks and their order are unknown, reflecting practical cases. We assume that all the tasks are equally important, and each has a fixed and known training budget. The main aim is to train a solver capable of generating high-quality solutions across all the learned tasks arising over time. Therefore, after training on each new task, the solver is tested on all tasks learned so far.

We propose LLR-BC, inspired by the experience replay paradigm (Isele & Cosgun, 2018; Buzzega et al., 2020). It consolidates knowledge learned from previous tasks by revisiting the old experiences obtained from these tasks. To increase the effectiveness of utilizing limited old experiences, we introduce two ingredients in LLR-BC: CaEW and DsBC. Figure 2 illustrates LLR-BC's full workflow. LLR-BC maintains a fixed-size experience memory and trains the solver learned from the previous tasks on each newly arrived task. Following standard practice in neural VRP solvers, LLR-BC trains on each task in epochs. In each epoch, LLR-BC iteratively samples and solves a batch of problem instances $\mathcal{I}$ from the current task to obtain a set of experience trajectories $\{\tau\}$. A DRL algorithm that maximizes the reward gain based on $\{\tau\}$ is applied to update the solver model. Simultaneously, a subset of experiences $\mathcal{E}$ is sampled from the memory for consolidation of previously learned tasks. CaEW assigns higher weights to more crucial sampled experiences, then DsBC guides model updates by minimizing weighted behavioral divergence between the current model and the behavior buffer of the sampled experiences. This joint process mitigates catastrophic forgetting while preserving adaptability. Notably, LLR-BC is a general and readily integrable framework, independent of specific model architectures or RL algorithms, and applicable to many existing neural solvers, such as POMO (Kwon et al., 2020) and INViT (Fang et al., 2024). Core ingredients of LLR-BC are described below. More details about LLR-BC can be found in Appendix A.

### 3.1 EXPERIENCE REPLAY FOR ROUTING

LLR-BC employs reservoir sampling (Vitter, 1985) to maintain a fixed-size buffer $\mathcal{B}$. New experiences are added directly if the memory is not full. Otherwise, each incoming experience replaces a randomly selected buffered experience with probability $\frac{|\mathcal{B}|}{N}$, where $|\mathcal{B}|$ is the buffer size and $N$ denotes the total number of experiences tried to add so far. Reservoir sampling ensures that all experiences are equally likely to be buffered while the total number of obtained experiences is increasing.

While existing methods (Li et al., 2024; Feng et al., 2025) take one whole problem instance of previous tasks as an experience, we consider experiences at finer granularity. We follow the commonly studied Markov decision process (MDP) formulation of constructive neural VRP solver (Kwon et al., 2020; Zhou et al., 2023; Bogyrbayeva et al., 2024), where each state corresponds to a partial solution composed of visited nodes, and an action is to select one node to visit next. In LLR-BC, each experience $e = \langle s, \mathcal{P} \rangle$ consists of a state $s$ and the solver's behavior $\mathcal{P}$ on the state $s$. A state $s$ represents the current partial solution for solving an instance. The behavior $\mathcal{P}$ is the probability distribution of

selecting each node as the next visit. The detailed solver design, e.g., the encoding of problems and solutions, follows the underlying base neural solver used. We use the probability distribution over all nodes rather than the single selected node to represent the solver's behavior, as it captures richer information about the learned routing strategy (Rusu et al., 2016; Buzzega et al., 2020).

To consolidate high-quality behaviors, LLR-BC buffers experiences only during the final epoch of each task, when the solver is expected to be well-trained on the task. At each model update step, a random set of experiences $\mathcal{E}$ is sampled from buffer $\mathcal{B}$ for behavior consolidation. A detailed discussion of the additional memory usage from the buffer can be found in Appendix A.5.

## 3.2 CONFIDENCE-AWARE EXPERIENCE WEIGHTING (CaEW)

While experiences are sampled uniformly from the buffer to ensure broad coverage of seen states, their importance for addressing forgetting can vary significantly. Constructive VRP solvers select nodes in a sequential manner, hence each decision influences future decisions while some crucial ones have a cumulative and amplified impact on the solution quality (Sun et al., 2024). Therefore, identifying and prioritizing crucial states and behaviors, i.e., crucial experience, is essential for effective learning and consolidation.

Decisions made with low confidence could be more susceptible to change during model updates (Farahmand, 2011; Ahmed et al., 2019). Therefore, LLR-BC assigns higher consolidation weights to such experiences, encouraging the solver to preserve behaviors in crucial decision points. Confidence is measured by the variance of the action probability distribution, where lower variance indicates lower confidence and leads to greater emphasis during consolidation (Spielberg & Azaria, 2019; Balasuntharam et al., 2023). Confidence-aware weight of an experience $e = \langle s, \mathcal{P} \rangle$ is normalized by $w(e) = 1 - \frac{\text{var}(\mathcal{P})}{\text{var}_{\max}(|\mathcal{P}|)}$, where $\text{var}(\mathcal{P})$ is the variance of $\mathcal{P}$ and $|\mathcal{P}|$ is the size of $\mathcal{P}$, i.e., the number of actions. $\text{var}_{\max}(|\mathcal{P}|) = \frac{|\mathcal{P}|-1}{|\mathcal{P}|^2}$ denotes the maximum possible variance for a distribution with the same number of candidates as $\mathcal{P}$. Then, given a set of sampled experiences $\mathcal{E}$, we rescale weights such that the sum of weight equals to 1, i.e., $\bar{w}_{\mathcal{E}}(e) = \frac{w(e)}{\sum_{e' \in \mathcal{E}} w(e')}, \forall e \in \mathcal{E}$. Appendix A.4 provides further discussion about the design of CaEW.

## 3.3 DECISION-SEEKING BEHAVIOR CONSOLIDATION (DsBC)

Existing methods (Li et al., 2024; Feng et al., 2025) replay experiences by generating, solving, and learning instances of previous tasks, ignoring the instructive information contained in historical behaviors. In contrast, by minimizing the weighted sum of the difference between the current model's behaviors and the buffered behavior at buffered states, LLR-BC encourages the consolidation of prior behavior patterns, thereby preserving knowledge and mitigating catastrophic forgetting.

The Kullback–Leibler divergence (KLD) $D_{KL}(P||Q)$, where $P$ and $Q$ are the teacher and leaner probability distribution respectively, is widely adopted for knowledge distillation of neural VRP solvers (Bi et al., 2022; Li et al., 2024; Zheng et al., 2025). However, recent studies (Kaplanis et al., 2019; Wang et al., 2024a; Qi et al., 2025) have shown that using reverse KLD (RKLD) $D_{RKL}(P||Q) = D_{KL}(Q||P)$ (detailed explanation is in Appendix A.3), could lead to better learning performance. While minimizing KLD pushes the learner distribution to spread probability mass across all modes of the given teacher distribution, minimizing RKLD leads to mode-seeking, where the learner tends to concentrate on the teacher's highest-probability actions, while still maintaining relatively good alignment with the overall distribution (Chan et al., 2022; Wu et al., 2025). During problem-solving of constructive neural VRP solvers, the action (node) with the highest probability is typically chosen to extend the route. To address forgetting, that could manifest in significantly longer routes on previously seen tasks, it is crucial to preserve these highest-probability decisions. Therefore, instead of KLD, LLR-BC employs the RKLD to measure the behavioral difference in a decision-seeking way. To consolidate sampled behaviors, LLR-BC minimizes the following behavior consolidation loss term, considering the RKLD (Wang et al., 2024a; Qi et al., 2025):

$$\mathcal{L}_{BC}(\theta, \mathcal{E}) = \sum_{e \in \mathcal{E}} \bar{w}_{\mathcal{E}}(e) \sum_{a \in \mathcal{A}} \mathcal{P}_\theta(a) \log \frac{\mathcal{P}_\theta(a)}{\mathcal{P}(a)}, \tag{1}$$

where $\mathcal{E} = \{e = \langle s, \mathcal{P} \rangle\}$ is the set of sampled experiences, $\mathcal{A}$ is the action space (i.e., available node set), $\mathcal{P}(a)$ is the probability corresponding to action $a$ in the buffered behavior $\mathcal{P}$ and $\mathcal{P}_\theta(a) = \pi_\theta(a|s)$ is the probability output by the current model $\pi_\theta$ on buffered state $s$.

Finally, during each model updating in training on a new task, LLR-BC jointly maximizes the expected reward on the new task and minimizes the RKLD between the current policy's behavior on the buffered states and the corresponding buffered behavior. The overall loss function is as follows:

$$\mathcal{L}(\theta, \{\tau\}, \mathcal{E}) = \mathcal{L}_{DRL}(\theta, \{\tau\}) + \alpha \cdot \mathcal{L}_{BC}(\theta, \mathcal{E}), \qquad (2)$$

where $\{\tau\}$ is the set of newly obtained experience trajectories from the new task, $\mathcal{L}_{DRL}(\theta, \{\tau\})$ is the loss calculated by the adopted DRL algorithm based on $\{\tau\}$ to maximize the reward gain, and $\alpha$ is a hyperparameter to balance between consolidation of previous experiences and learning of the new task. Notably, LLR-BC is a generic framework and can be applied to different neural solver methods with different DRL algorithms, in which $\mathcal{L}_{DRL}$ in Eq. equation 2 can be implemented in different ways. Appendix A.5 detailedly discusses about computational cost of DsBC.

## 4 EXPERIMENTS

We conduct a series of experiments to answer the following research questions:

- **Effectiveness on all learned tasks:** Can LLR-BC effectively solve learned tasks after training on them sequentially in the lifelong learning setting?

- **Stability and plasticity:** In lifelong learning process, how does LLR-BC perform in terms of stability and plasticity?

- **Zero-shot generalization ability:** Can LLR-BC effectively acquire transferable knowledge across sequentially arising tasks to enhance zero-shot performance on an unseen task?

- **Hyperparameter sensitivity:** How sensitive is LLR-BC to key hyperparameters, including the buffer size $|\mathcal{B}|$, number of sampled experiences $|\mathcal{E}|$, and the weight $\alpha$ of the behavior consolidation term in the loss function?

- **Applicability:** Can LLR-BC work effectively on different base neural solvers?

To answer the questions, we simulate multiple lifelong learning scenarios on CVRP and TSP, and compare LLR-BC against widely used and representative lifelong learning baselines across diverse metrics[1]. Most experiments are conducted based on POMO (Kwon et al., 2020), given its concise design and broad applicability. To verify the applicability of LLR-BC, we further evaluate LLR-BC on two representative and state-of-the-art constructive neural solvers designed for good cross-distribution cross-scale performance, i.e., Omni (Zhou et al., 2023) and INViT (Fang et al., 2024).

**Dataset.** Six distributions for sampling node coordinates are used: four proposed by Bossek et al. (2019) and widely used in neural VRP solver studies (Jiang et al., 2022; Zhou et al., 2023), i.e., Uniform (U), Gaussian Mixture (GM), Explosion (E), Compression (C), and two additionally designed ones for greater diversity: Grid (G) and Ring (R). For CVRP, node demand is also required to be sampled. While prior studies used uniformly random demand for all distributions (Kwon et al., 2020; Jiang et al., 2022; Zhou et al., 2023; Fang et al., 2024), we build six distinct demand distributions and assign them to the above six distributions to better reflect the diversity of real-world problems (Liu et al., 2021). Following common practice (Kwon et al., 2020; Jiang et al., 2022), we consider three problem scales: 20, 50, and 100. Specifically, tasks U and R correspond to scale 20, tasks G and E to scale 50, and tasks C and GM to scale 100, arbitrarily. We construct five task orders by randomly permuting the tasks. In addition, two classic and widely used benchmark datasets, TSPLIB (Reinelt, 1991) and CVRPLIB (Uchoa et al., 2017), are used for generalization ability evaluation. Appendix B provides the full definition of tasks and orders.

**Training and Test Settings.** Following Kwon et al. (2020), we train the solver on each task over 200 epochs, and solve a batch of sampled problems in parallel to leverage GPU parallelism during training. Therefore, both experience buffering and sampling operate at the batch level rather than the single experience level. Hyperparameters of LLR-BC are set as follows: $|\mathcal{B}| = 1000$, $|\mathcal{E}| = 16$,

---

[1]Our implementation is available in https://github.com/PeiJY/LLR-BC.

and $\alpha = 100$, where $|\mathcal{B}|$ and $|\mathcal{E}|$ are defined in units of experience batches. The buffer constitutes only about 0.01% of the total training experiences obtained across all tasks. For evaluation, each task has a test set of 1,000 problem instances, which are different from the training ones. For each trial, all methods are evaluated on the same test sets. Additional details are provided in Appendix C.

**Compared Methods.** The goal of this work is not to design a totally new solver that outperforms the state-of-the-art solver on each individual task or a fixed set of tasks that can be learned simultaneously, but to improve the solver's ability to learn unpredictable tasks sequentially. We consider several existing methods for comparison. First, we adapt the methods of (i) Li et al. (2024) (denoted Li (intra) and Li (inter)) and (ii) of Feng et al. (2025) (denoted Feng) to our scenarios and include them as baselines. Then, we involve several representative and commonly adopted strategies in neural solver studies (Kwon et al., 2020; Zhou et al., 2023): (iii) *Restart*, reinitialize the solver and train from scratch on each task; (iv) *Fine-tuning*, simply sequential training on each task. Additionally, we evaluate (v) EWC (Kirkpatrick et al., 2017), a widely used method in lifelong learning, especially in recent studies of optimization tasks (Manchanda & Ranu, 2023; Pei et al., 2025a). We also adapt and compare with (vi) LiBOG (Pei et al., 2025a), a recent method originally proposed for black-box optimization, within the VRP tasks. Appendix C.1 provides more details.

**Evaluation Metrics.** Based on a standard practice in lifelong learning research (Chaudhry et al., 2018; Wang et al., 2024c), we build the following evaluation metrics. Let $d_{i,j}$ denote the test performance, i.e., average objective value (tour length) over all the test instances, of the solver on task $T_j$ after training on the first $i$ tasks sequentially (including task $T_i$). As different tasks have different scales for the objective value, the objective values are normalized by $\bar{d}_{i,j} = \frac{d_{i,j}-d_j^*}{d_j^*}$, where $d_j^*$ is smallest (best) test performance achieved on task $T_j$ by all the solvers obtained by all the methods in all lifelong learning scenarios. Based on this notation, after learning $k$ tasks, we calculate:

- **Average Performance (AP)**: the current average performance on tasks learnt so far, i.e., $AP = \frac{1}{k} \sum_{i=1}^{k} \bar{d}_{k,i}$.

- **Average Forgetting (AF)**: the average performance decrease on previously learned tasks after learning all of them sequentially, i.e., $AF = \frac{1}{k-1} \sum_{i=1}^{k-1} \max(0, \bar{d}_{k,i} - \bar{d}_{i,i})$.

- **Average Max Forgetting (AMF)**: the average of maximal forgetting of each task during the lifelong learning, i.e., $AMF = \frac{1}{k-1} \sum_{i=1}^{k-1} \max_{j=i+1}^{k} \max(0, \bar{d}_{j,i} - \bar{d}_{i,i})$.

- **Average Plasticity (APl)**: the average performance a solver achieves on each new task after training on it, i.e., $APl = \frac{1}{k} \sum_{i=1}^{k} \bar{d}_{i,i}$.

- **Average zero-shot Generalization (AG)**: the average performance on each newly arising task before training on it, i.e., $AG = \frac{1}{k-1} \sum_{i=1}^{k-1} \bar{d}_{i,i+1}$.

Smaller values of the above metrics indicate better performance. Unless otherwise specified, all reported values of the five metrics in tables are scaled by $10^{-3}$.

## 4.1 Performance in Solving Seen Tasks

The experimental results for CVRP and TSP are summarized in Table 1, with $k = 6$ for all metrics. More detailed results with the absolute solution distance and optimality gap can be found in Appendices D.1, D.2. Across all task orders, LLR-BC achieves average AP values of $0.0042$ (CVRP) and $0.0034$ (TSP), whereas all other methods exceed $0.023$ (CVRP) and $0.014$ (TSP), respectively, indicating that LLR-BC obtains solutions of problems from learned tasks with substantially better quality. Moreover, LLR-BC exhibits greater robustness to task order variation. Compared methods show large fluctuations (larger Std.) in AP across orders, while LLR-BC consistently maintains low AP values with lower standard deviations. An additional comparison of LLR-BC with methods with non-lifelong settings, including POMO in multi-task training settings and INViT in its original training setting (Fang et al., 2024), further demonstrates LLR-BC's superior performance (details can be found in Appendix D.3). These results demonstrate that LLR-BC effectively learns from sequentially arising tasks and retains the ability to solve all encountered tasks with high effectiveness.

Table 1: The mean (std.) of metrics over the five task orders.

| Method | CVRP | | | | | TSP | | | | |
|---|---|---|---|---|---|---|---|---|---|---|
| | AP | AF | AMF | APl | AG | AP | AF | AMF | APl | AG |
| Li (inter) | 32.0 (6.8) | **0.0 (0.0)** | **0.0 (0.0)** | 33.6 (7.7) | 40.1 (10.5) | 56.5 (38.3) | 0.4 (0.5) | **0.5** (0.5) | 61.7 (45.6) | 77.6 (59.6) |
| Li (intra) | 34.1 (2.1) | **0.0 (0.0)** | 0.1 (0.1) | 39.3 (1.7) | 48.2 (4.8) | 54.2 (7.2) | **0.2 (0.3)** | **0.5** (0.4) | 62.8 (12.5) | 82.4 (27.7) |
| Feng | 24.6 (2.0) | 3.2 (2.0) | 4.5 (1.7) | 24.2 (**0.7**) | 39.1 (8.6) | 24.1 (3.2) | 1.8 (1.4) | 3.8 (1.9) | 21.4 (1.6) | 45.5 (23.2) |
| Restart | 60.5 (29.3) | 41.3 (26.3) | 52.0 (25.5) | 9.1 (-) | 49.5 (4.7) | 31.7 (7.6) | 50.5 (40.0) | 65.6 (32.8) | 7.1 (-) | 72.4 (**9.2**) |
| Fine-tuning | 23.5 (9.2) | 19.9 (3.6) | 28.1 (5.0) | 3.8 (0.8) | 42.1 (6.7) | 14.8 (2.3) | 28.9 (10.9) | 36.6 (7.4) | 3.5 (**1.3**) | 57.4 (17.7) |
| EWC | 28.3 (9.2) | 19.5 (5.4) | 25.5 (4.3) | 6.9 (1.1) | 39.8 (4.2) | 18.3 (2.8) | 18.6 (4.4) | 24.6 (5.5) | 5.5 (1.7) | 52.1 (18.3) |
| LiBOG | 31.3 (11.9) | 19.7 (5.8) | 25.1 (5.8) | 7.2 (1.0) | 40.8 (**3.9**) | 19.2 (1.6) | 17.2 (4.5) | 22.8 (5.5) | 5.8 (1.7) | 51.7 (14.8) |
| **LLR-BC** | **4.2 (1.0)** | 0.7 (0.5) | 0.8 (0.4) | **3.5 (0.8)** | **26.7** (4.5) | **3.4 (1.3)** | 0.8 (0.3) | 1.1 (**0.4**) | **2.8** (1.6) | **41.1** (21.0) |

(a) CVRP   (b) TSP

Figure 3: Forgetting curve of task order 1, measured by average solution distance (vertical axis). Epochs 0–200 (first task) are omitted as no forgetting occurs. Notably, some methods obtain too large solution distances and exceed the vertical range.

## 4.2 STABILITY AND PLASTICITY

We analyze the stability by comparing the AF and AMF values over all five task orders, as shown in Table 1. Li (inter) and Li (intra) (Li et al., 2024), perform better in mitigating forgetting than LLR-BC. This is because half of their training resources (in terms of epochs) on the new task are used to learn purely from instances of previous tasks. Although they address forgetting well, they do not learn much from new tasks and perform poorly in terms of AP and APl. LLR-BC achieves substantially lower AF and AMF values than the rest of the compared methods, demonstrating its strong stability for effectively mitigating catastrophic forgetting. Figure 3 shows the average solution distance on seen tasks during lifelong learning of task order 1. At the start of each task, most baselines exhibit a sharp performance drop on previously learned tasks. Li (inter) and Li (intra) learn new tasks slowly. Feng suffers from low stability in maintaining learned knowledge. In contrast, LLR-BC mitigates forgetting and maintains consistently lower distances with greater stability.

LLR-BC also achieves the best APl values among all methods, indicating its best plasticity. Notably, *Restart* produces a single APl value with no std., as it is independent of task order and is executed only once, rather than once per order. Figure 4 presents the solver's test performance on the current task throughout lifelong learning. As the number of encountered tasks increases, EWC, as well as LiBOG, gradually loses plasticity due to the cumulative regularization on model parameters. In contrast, LLR-BC operates at the behavioral level, enabling the model to explore the parameter space more freely while preserving learned behaviors. Model can discover new parameter values that yield good decisions on both new and past tasks. With the guidance from previous experiences, LLR-BC demonstrates even better performance on the new task than *fine-tuning*.

Additionally, the rankings of methods differ on the two forgetting metrics, i.e., AF and AMF. For example, *fine-tuning* achieves a lower (better) AF than EWC but yields a higher (worse) AMF.

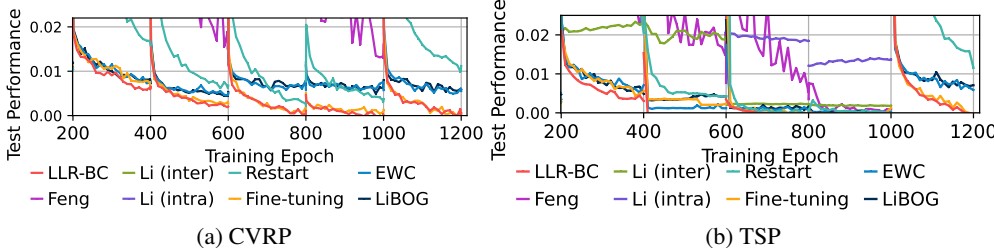

Figure 4: Test performance on the current task during lifelong learning on task order 1.

This discrepancy arises because AF also accounts for backward transfer, i.e., training on a new task can sometimes improve performance on earlier, similar tasks. Backward transfer is also evident in Figure 3. This further underscores the importance of LLR-BC's ability to maintain strong plasticity. Similar patterns are observed across other task orders, as shown in Appendix D. In summary, LLR-BC consistently outperforms all baselines in both stability and plasticity.

## 4.3 ZERO-SHOT GENERALIZATION

Beyond performance on learned tasks, we also expect neural solvers to acquire general knowledge from randomly ordered tasks, thereby improving their performance on unseen tasks. AG values for all methods are reported in Table 1. LLR-BC outperforms all baselines in terms of AG, demonstrating its strong ability to zero-shot generalize to a new task. We further test the solver after learning from task order 1 on TSPLIB and CVRPLIB, with the maximum scale as 1001. As shown by the results in Table 2, LLR-BC performs the best on both benchmarks. Appendix D.4 presents the detailed values. Despite the uncontrollable and unpredictable nature of task order, LLR-BC demonstrates a strong generalization ability by capturing general knowledge across tasks.

Table 2: Mean (Std.) test performance on benchmark instances.

| Benchmark | Restart | Fine-tuning | EWC | LiBOG |
|---|---|---|---|---|
| CVRPLIB | 43.68 (27.37) | 8.54 (10.71) | 11.86 (11.59) | 18.05 (10.86) |
| TSPLIB | 99.76 (61.06) | 38.16 (93.36) | 27.80 (59.58) | 27.96 (32.29) |

| Benchmark | Li (inter) | Li (intra) | Feng | **LLR-BC** |
|---|---|---|---|---|
| CVRPLIB | 39.31 (23.11) | 46.80 (26.96) | 27.38 (18.53) | **7.88 (8.75)** |
| TSPLIB | 122.60 (51.78) | 58.46 (26.55) | 75.15 (30.83) | **18.08 (16.98)** |

## 4.4 ABLATION STUDY

We conduct ablation studies (denoted -nEW and -KLD) on task order 1 to evaluate the contribution of our key components, i.e. CaEW and DsBC. As shown in Table 3, removing either key component leads to performance degradation across multiple metrics on both CVRP and TSP, confirming the effectiveness and contribution of both modules. Furthermore, we conduct additional ablation studies to better investigate the characteristics of LLR-BC, including: (i) a variant that buffers experiences at every epoch instead of only the last epoch of each task (denoted -EE), (ii) variants that use entropy and top-2 margin instead of variance to quantify decision confidence (denoted -Ent and -T2M), (iii) a variant that rescales the reservoir sampling probabilities so that the buffered experiences are more uniformly distributed across tasks (denoted -Res), and (iv) a variant that buffers all steps for solve an instance as one experience, instead of just one step, denoted as -IB. The results show that both buffering only at the final epoch and using unscaled reservoir sampling contribute notably to the performance of LLR-BC. Replacing the confidence measure has only a minor impact, indicating that LLR-BC is not sensitive to the specific choice of confidence measurement. More details about ablation studies can be found in Appendix C.3.

## 4.5 HYPERPARAMETER SENSITIVITY ANALYSIS

We varied the values of key hyperparameters, i.e., $|\mathcal{E}|$, $|\mathcal{B}|$ and $\alpha$, and evaluated the performance of LLR-BC across all metrics under each setting, as shown in Table 3. *-nEW* and *-KLD* denote the version without CaEW (using equal weights) and the version using KLD, respectively. *-n(v)*

denotes the version with $n$ set to $v$. Detailed discussion can be found in Appendix D.5. In summary, varying the hyperparameters leads to only minor performance fluctuations across evaluation metrics, compared with the performance gap between LLR-BC and the baselines. This suggests that LLR-BC is robust and not overly sensitive to its hyperparameter settings.

Table 3: Metric values of LLR-BC in different settings, on task order 1 with $k = 6$.

| Method | CVRP | | | | | TSP | | | | | Method | CVRP | | | | | TSP | | | | |
|---|---|---|---|---|---|---|---|---|---|---|---|---|---|---|---|---|---|---|---|---|---|
| | AP | AF | AMF | APl | AG | AP | AF | AMF | APl | AG | | AP | AF | AMF | APl | AG | AP | AF | AMF | APl | AG |
| -$\mathcal{E}$(4) | 5.7 | 0.8 | 0.8 | 4.8 | 22.7 | 2.6 | 1.2 | 1.2 | 1.7 | 21.6 | -$\alpha$(10) | 13.2 | 5.1 | 5.5 | 4.2 | 34.7 | 1.3 | 0.4 | 0.6 | 1.2 | 20.5 |
| -$\mathcal{E}$(8) | 5.8 | 1.0 | 1.1 | 4.2 | 22.8 | 2.0 | 0.9 | 1.1 | 1.5 | 21.6 | -$\alpha$(50) | 6.6 | 1.4 | 1.6 | 4.2 | 24.7 | 2.0 | 0.7 | 0.9 | 1.2 | 22.2 |
| -$\mathcal{B}$(250) | 6.4 | 1.9 | 1.9 | 4.3 | 25.3 | 1.6 | 0.7 | 1.1 | 1.1 | 20.5 | -$\alpha$(500) | 5.1 | 0.2 | 0.3 | 4.9 | 22.5 | 3.5 | 1.3 | 1.3 | 3.2 | 21.2 |
| -$\mathcal{B}$(500) | 5.8 | 1.3 | 1.3 | 4.2 | 29.8 | 2.2 | 0.6 | 0.9 | 1.7 | 22.2 | -$\alpha$(1000) | 5.9 | 0.3 | 0.4 | 5.8 | 23.6 | 4.9 | 1.4 | 1.4 | 4.6 | 22.3 |
| -nEW | 5.2 | 0.8 | 0.8 | 4.2 | 24.5 | 1.8 | 0.9 | 0.8 | 1.5 | 22.3 | -KLD | 5.5 | 0.7 | 0.7 | 4.3 | 23.1 | 1.9 | 0.9 | 0.9 | 1.5 | 22.3 |
| -EE | 7.8 | 3.1 | 3.1 | 3.9 | 24.1 | 2.6 | 2.1 | 2.4 | 1.6 | 22.7 | -Ent | 4.8 | 0.5 | 0.7 | 4.0 | 22.7 | 2.1 | 0.9 | 1.0 | 1.6 | 21.9 |
| -Res | 4.9 | 0.9 | 0.9 | 3.7 | 23.9 | 2.0 | 1.0 | 1.0 | 1.3 | 22.3 | -T2M | 4.8 | 1.0 | 1.2 | 3.4 | 22.8 | 1.7 | 0.9 | 1.1 | 1.0 | 21.6 |
| **Default** | 4.9 | 0.6 | 0.7 | 4.3 | 23.5 | 1.7 | 0.8 | 0.9 | 1.3 | 21.6 | -IB | 35.4 | 23.4 | 27.2 | 4.4 | 31.0 | 2.5 | 1.8 | 2.0 | 1.0 | 21.9 |

## 4.6 APPLICABILITY

We further implement LLR-BC on Omni Zhou et al. (2023) and INViT Fang et al. (2024), and compare it with *fine-tuning* on task order 1. As Table 4 shows, LLR-BC outperforms *fine-tuning* generally, with a similar pattern found on POMO. More details of experiment settings, results, and discussions are presented in Appendix D.6. In summary, LLR-BC is effective on the focused lifelong learning scenario, without relying on specific characteristics of the base neural solver.

Table 4: Metric values with Omni or INViT as the base neural solver, $k = 6$.

| Method | CVRP | | | | | TSP | | | | |
|---|---|---|---|---|---|---|---|---|---|---|
| | AP | AF | AMF | APl | AG | AP | AF | AMF | APl | AG |
| Fine-tuning (Omni) | 34.7 | 19 | 22.1 | **16.9** | 64 | 52.9 | 13.6 | 18.8 | 14.4 | 56.7 |
| **LLR-BC (Omni)** | **16.5** | **2.9** | **4.6** | 19.9 | **55.2** | **12.9** | **0.9** | **1.4** | **11.6** | **34.7** |
| Fine-tuning (INViT) | 28.6 | 9.7 | 9.7 | 21.6 | 35.2 | **14.5** | **0.6** | **0.6** | 17.1 | 11.1 |
| **LLR-BC (INViT)** | **23.8** | **5.8** | **5.8** | **19.6** | **28.8** | **14.5** | 4.5 | 6.3 | **11.1** | **10.7** |

## 5 CONCLUSIONS

We propose LLR-BC, a lifelong learning framework for neural VRP solvers, considering learning tasks with varying problem scales and distributions that arrive sequentially. We introduce two core components: CaEW, which emphasizes low-confidence behaviors, and DsBC, which preserves learned behavior effectively in a decision-seeking way. Experiments across diverse orders of tasks with varying distributions and scales demonstrate that LLR-BC consistently outperforms baselines in terms of performance in solving learned tasks, resistance to catastrophic forgetting, learning efficiency on new tasks, and zero-shot generalization. Ablation and sensitivity studies confirm the effectiveness of the core designs of LLR-BC and the framework's robustness to hyperparameter settings. LLR-BC is both model- and RL algorithm-agnostic, and can be integrated with various neural VRP solvers. Experiments on different base neural solvers verify LLR-BC's applicability.

Although studied focus on cross-distribution and cross-scale settings, LLR-BC is in principle applicable to other lifelong learning scenarios for neural VRP solvers, such as tasks that differ in their distance matrices. Lifelong learning across different problem variants (e.g., TSP and CVRP) is also a practically important research direction, which we leave for future work, although it may require task-specific model components (Drakulic et al., 2025). LLR-BC could further reduce its reliance on task identity and adapt to scenarios with continuously evolving tasks by applying reservoir sampling during training on each instance, rather than only at the final epoch of each task. Despite its strong performance, LLR-BC has limitations in certain cases. For example, a fixed $|\mathcal{E}|$ can be difficult to tune when task scales differ substantially, potentially leading to reduced plasticity on small-scale tasks, where old experiences dominate new ones in a batch, and reduced stability on large-scale tasks, where new experiences dominate old ones in a batch. Addressing these issues in future work could further broaden the applicability of LLR-BC.

ACKNOWLEDGEMENT

This work was supported in part by the New Zealand MBIE Endeavour Smart Ideas Grant under Contract RTVU2305 and MBIE SSIF Fund on Data Science Programme under contract RTVU1914. Jialin Liu and Xin Yao were supported by internal grants from Lingnan University, Hong Kong SAR, China.

REPRODUCIBILITY STATEMENT

All algorithmic details (cf. Section 3 and Appendix A), training protocols (cf. Section 4 and Appendix C), and evaluation metrics (cf. Section 4 and Appendix C.2) are described in the main paper and further elaborated in the Appendix. For empirical studies, we provide a detailed description of the datasets and instance generation pipeline (cf. Appendix B). Hyperparameters and implementation details for all baselines are also reported in Section 4 and Appendix C. We also release our code and scripts for reproducing our experiments, including instructions for running and data preparation. Together, these resources enable independent researchers to replicate our results and build upon our contributions.

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

# A  FURTHER DETAILS OF LLR-BC

## A.1  OVERALL PROCESS OF LLR-BC

For a sequence of tasks, LLR-BC conducts lifelong learning with Algorithm 1. For each new task in the lifelong learning process, LLR-BC trains the solver model $\pi_\theta$ and updates the experience buffer $\mathcal{B}$ as Algorithm 2 demonstrates. Only the model parameter $\theta$ and the buffer $\mathcal{B}$ are transferred between tasks. LLR-BC can sequentially train on any number of tasks, as it does not rely on any knowledge about the number or order of tasks.

---

**Algorithm 1** LLR-BC Lifelong Learning

---

1: **Input:** $\{T_i\}_{i=1}^K, \theta_0$
2: **Parameters:** $A, C, \mathcal{L}_{DRL}$
3: **Output:** $\theta$
4: **for** $i \in \{1, \ldots, K\}$ **do**
5: $\quad \theta, \mathcal{B} \leftarrow$ LLR-BC One Task Learning $(\theta, \mathcal{B}, T, A, C, \mathcal{L}_{DRL})$
6: **end for**
7: **return** $\theta$

---

**Algorithm 2** LLR-BC One Task Learning

---

1: **Input:** $\theta, \mathcal{B}, T, A, C, \mathcal{L}_{DRL}$
2: **Parameters:** $\alpha, |\mathcal{E}|$
3: **Output:** $\theta, \mathcal{B}$
4: **for** $i \in \{1, \ldots, A\}$ **do**
5: $\quad \mathcal{I} \leftarrow$ generate a batch of $C$ problems from $T$
6: $\quad \{\tau\} \leftarrow$ Solve $\mathcal{I}$ with $\pi_\theta$
7: $\quad loss \leftarrow \mathcal{L}_{DRL}(\theta, \{\tau\})$
8: $\quad$ **if** $\mathcal{B}$ is not empty **then**
9: $\quad\quad \mathcal{E} \leftarrow$ uniformly sample
10: $\quad\quad \{w_\mathcal{E}(e)\}_{e \in \mathcal{E}} \leftarrow \{1 - \frac{\text{var}(\mathcal{P})}{\text{var}_{\max}(\mathcal{P})}\}_{e = \langle s, \mathcal{P} \rangle \in \mathcal{E}}$
11: $\quad\quad \{\bar{w}_\mathcal{E}(e)\}_{e \in \mathcal{E}} \leftarrow \{\frac{w_\mathcal{E}(e)}{\sum_{e' \in \mathcal{E}} w_\mathcal{E}(e')}\}_{e \in \mathcal{E}}$
12: $\quad\quad loss \leftarrow loss + \alpha \cdot \mathcal{L}_{BC}(\theta, \mathcal{E})$
13: $\quad$ **end if**
14: $\quad \theta \leftarrow$ Optimize$(\theta, loss)$
15: $\quad$ **if** $i = A$ **then**
16: $\quad\quad \mathcal{B} \leftarrow$ reservoir sampling $(\{\tau\}, \mathcal{B})$
17: $\quad$ **end if**
18: **end for**
19: **return** $\theta, \mathcal{B}$

---

## A.2  CONFIDENCE IN CONSTRUCTIVE SOLVERS

Figure 5 demonstrates a CVRP example where the low-confidence decision changes when training the solver on a new task, For the solver trained on U, tested on a problem of task U, it generates a solution with some low-confidence decisions, e.g., the 5th action. Then, after fine-tuning it on task E for 10 epochs, we test it again on the same problem of task U, getting a new solution (the total distance is larger). On the node corresponding to the previous 5th decision, the new solution differs from the previous one. The 11th actions are also changed, where the new solver has low confidence. Many decisions after are also different, but it could be due to the previous differences. It illustrates that low-confidence decisions could be more likely to drift during training on a new task.

## A.3  REVERSE KLD

The Kullback–Leibler divergence (KLD) measures the difference from one learner probability distribution $Q$ to a target/teacher distribution $P$ and is widely adopted for knowledge distillation of

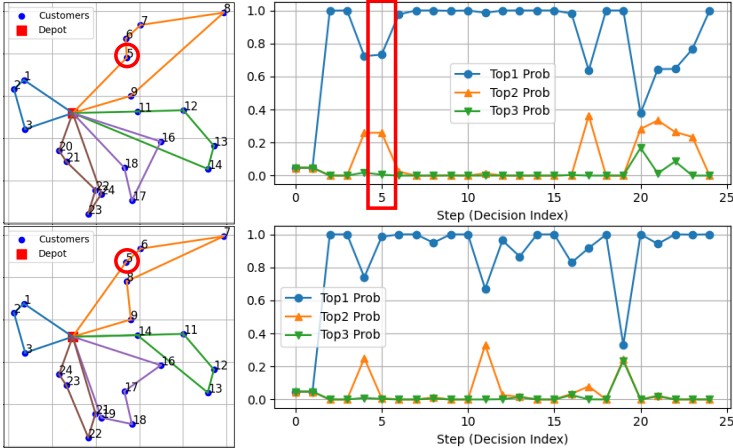

Figure 5: Example of decision drift in a node with low confidence. The number on a node indicates its order in the solution. Left: the generated solution on a problem of task U by a given solver. Right: the top three values of action probability from the solver corresponding to the generated solution. Upper: a solver trained on U task. Lower: fine-tuning the solver on task E for 10 epochs after training it on task U.

neural VRP solvers (Bi et al., 2022; Zheng et al., 2025), computed as:

$$D_{KL}(P||Q) = \sum_i P(i) \log \frac{P(i)}{Q(i)}, \tag{3}$$

where $P(i)$ and $Q(i)$ are the probability of candidate $i$ of the target/teacher distribution $P$ and of another distribution $Q$. KLD is asymmetric, i.e., $D_{KL}(P||Q) \neq D_{KL}(Q||P)$, although both attain their minimum when $P = Q$. By switching $Q$ and $P$ KLD convert to RKLD, i.e.,

$$D_{RKL}(P||Q) = D_{KL}(Q||P) = \sum_i Q(i) \log \frac{Q(i)}{P(i)}. \tag{4}$$

When the teacher's action probability $P(i)$ is close to zero, the KLD for action $i$ becomes very small. As a result, using KLD as a loss term places little emphasis on the mismatch between $P(i)$ and the learner's probability $Q(i)$ in such cases. This leads to overly mild penalties when the learner assigns high probability to actions that the teacher has effectively ruled out, i.e., learned incorrect actions. Consequently, the learner tends to spread probability mass across all actions to cover possible modes (Chan et al., 2022; Wu et al., 2025).

In contrast, RKLD places greater emphasis on penalizing mismatches when the learner's probability $Q(i)$ is high. This makes RKLD more effective at discouraging the learner from assigning high probability to those learned incorrect actions, thereby encouraging focus on the most probable teacher actions. As a result, RKLD promotes *mode-seeking* behavior (Chan et al., 2022; Wu et al., 2025).

In LLR-BC, the teacher $P$ is the buffered behavior, and the learner $Q$ is the behavior of the current learning solver on the corresponding buffered state. By minimizing $D_{RKL}(P||Q)$, LLR-BC better preserves the learned good decisions of previously learned tasks.

## A.4 CONFIDENCE-AWARE WEIGHTING

A low-confidence decision of the solver can arise in two situations: (i) there is a single best action, but the model has not fully learned the current state and therefore exhibits high uncertainty; or (ii) there are multiple equally good actions. Since we only buffer experiences at the final epoch of each task, when the model is already well trained on that task, both types of low-confidence decisions deserve higher emphasis (larger weights): (i) high uncertainty indicates that the state is difficult to learn and thus crucial for learning the task [3,4]; and (ii) multiple equally good actions,

which mostly lead to similarly good solutions, create plateaus in the solution space and make the optimization problem harder to solve.

CaEW is designed to assign smaller weights to steps where the model selects the estimated best action with higher probability (i.e., higher decision confidence). Besides variance, other probability-based confidence measures with similar properties could also be used. CaEW is not specialized for any particular confidence measure. We choose variance because it is simple, straightforward, and effective.

### A.5 COMPUTATIONAL COST

Importantly, compared with fine-tuning the base neural solver, LLR-BC only increases computational cost during training and does not incur extra overhead during inference.

**Memory** Compared with fine-tuning, the additional space complexity introduced by LLR-BC primarily arises from the experience buffer storing old experiences, with space complexity as $O(|\mathcal{B}| * s_e)$, with $s_e$ denotes the space for one unit of experience. $s_e$ could vary depending on the formulation of the solving process of the base neural solver. Under our implementation, with $|\mathcal{B}| = 1000$, the experience buffer occupies only about 400 MB of memory when solving CVRP with POMO as the base neural solver. It is acceptable considering the superior performance of LLR-BC.

**Run Time** Typically, in a training batch of problems with scale $N$, fine-tuning constraints at least $N$ steps of model forward propagation to generate complete solutions. Compared with fine-tuning the base neural solver, LLR-BC introduces additional time cost from computing the new behaviors of the current model on $|\mathcal{E}|$ sampled old states, storing and sampling experiences from the buffer, and calculating the RKLD between new and old behaviors as the LBC. The first part requires $|\mathcal{E}|$ additional forward propagations, while the latter two involve only lightweight arithmetic operations, whose run cost is negligible. Therefore, the extra time cost introduced by LLR-BC per problem batch is $O(|\mathcal{E}| * t_m)$, $|\mathcal{E}| * t_m$, with $t_m$ denotes the complexity of forward propagation. With $|\mathcal{E}| = 4$ (already outperforming the compared methods) and $N = 20, 50, 100$, ideally, LLR-BC incurs only an additional computational overhead of less than 20%, 8%, and 4%, respectively, compared with fine-tuning. Under our implementation, for all experiments, LLR-BC introduces no more than 8 additional hours based on *fine-tuning* of training for the whole lifelong learning process. Given its superior performance, this additional training complexity is a reasonable and acceptable trade-off.

## B TASK AND ORDER DETAILS

### B.1 TASK SETTINGS

Four of the six tasks use the node coordinate distribution defined in Bossek et al. (2019), which is widely used in neural VRP solver studies (Zhou et al., 2023; Bi et al., 2022). And we built two more distributions, i.e., Ring and Grid. We also assign different demand distributions to different tasks. The details of node coordinate distribution and demand distribution of each task are as follows. Vehicle capacity is set to 30, 40, 50 for the tasks with scale 20, scale 50, and scale 100, following the setting in Kwon et al. (2020). Notably, for CVRP, the coordinate of the depot is randomly sampled from $[0, 1]^2$, independent of the sampling of customers (nodes).

**Task Uniform (U):** Each customer/city node $i$ has coordinates $(x_i, y_i) \sim \mathcal{U}(0, 1)$. Demand $d_i \sim \mathcal{U}(1, 10)$, drawn uniformly for CVRP. Figure 6 demonstrates problem examples of task U on CVRP and TSP.

**Task Gaussian Mixture (GM):** 5 cluster centers $\{c_z\}_{z=1}^5$ are first uniformly sampled from $[0, 50]^2$. For each $c_z$ center, 19 customer/city nodes are sampled. Each is drawn as $(x_i, y_i) \sim \mathcal{N}(c_z, 1)$. Then all node coordinates are linearly mapped into $[0, 1]^2$ with minmax normalization. Demand of each center is drawn uniformly, $d_{c_z} \sim \mathcal{U}(1, 10)$. The distance $dist_i$ of each node $i$ to its center is calculated and minmax normalized to [0,1]. The demand of a node $i$ is set as $10 \cdot dist_i$, then we round the demand. Figure 7 demonstrates problem examples of task GM on CVRP and TSP.

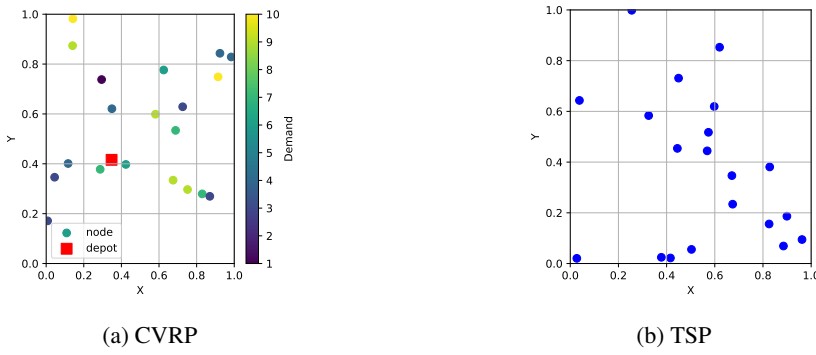

Figure 6: Problem instance sample of task U.

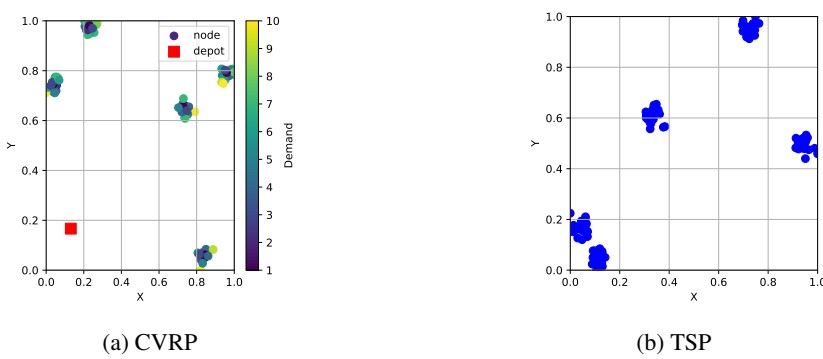

Figure 7: Problem instance sample of task GM.

**Task Explosion (E):** First nodes are sampled with a uniform coordinate distribution. Then, a point $p$ (not a node) is uniformly sampled from $[0, 1]^2$. For each node with distance to $p$ smaller than 0.3, we move he customer away from $p$ with moving length $0.3 + s, s \sim Exp(40)$. All nodes' coordinates are clamped into $[0, 1]^2$, finally. The demand of each node is sampled from $\mathcal{N}(5, 1)$ and then rounded and clamped to [0,10]. Figure 8 demonstrates problem examples of task E on CVRP and TSP.

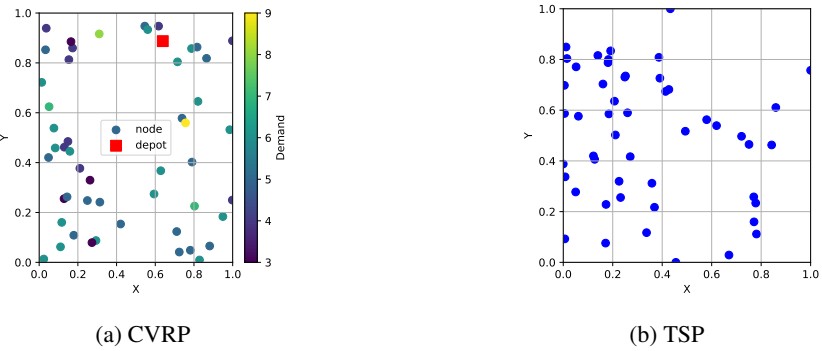

Figure 8: Problem instance sample of task E.

**Task Compression (C):** First nodes are sampled with a uniform coordinate distribution. Then 2 points $p_1$ and $p_2$ are sampled uniformly from $[0, 1]^2$, forming a line $l$. For each node with distance to the $l$ smaller than 0.3, a new distance is sampled from $\mathcal{N}(0, 0.1^2)$. The node will be along the direction vertical to the line, so that its distance to $l$ equals the new distance. All nodes' coordinates are clamped into $[0, 1]^2$, finally. The demand of each node is sampled as $10 - x, x \sim \mathcal{N}(5, 1)$, and

then rounded and clamped to [0,10]. Figure 9 demonstrates problem examples of task C on CVRP and TSP.

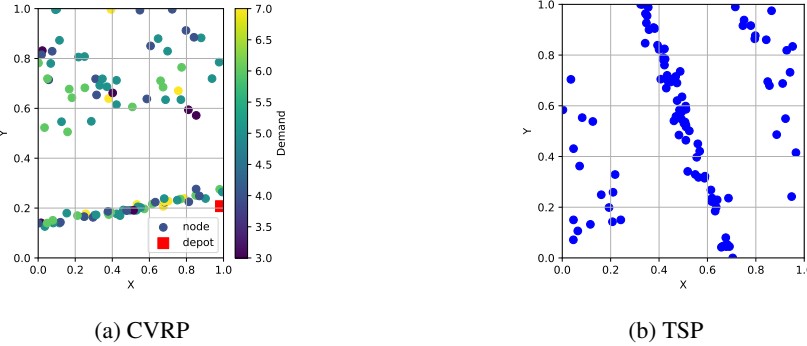

(a) CVRP

(b) TSP

Figure 9: Problem instance sample of task C.

**Task Grid (G):** A ratio $p$ is uniformly sampled from [0.2,0.8]. Sample a random number $r$ uniformly from [0,1]. If $r \leq 0.5$, $w = 1, h = p$, otherwise, $w = p, h = 1$. Sample a center rectangle $c$ (not node) $x \sim \mathcal{U}(\frac{w}{2}, 1 - \frac{w}{2})$,$y \sim \mathcal{U}(\frac{h}{2}, 1 - \frac{h}{2})$. Make a square grid in the rectangle with center as $c$, width as $w$ and height as $h$, $a = \lceil \sqrt{50 * \frac{w}{h}} \rceil$ grids in x axis direction, $b = \lceil \sqrt{\frac{50}{a}} \rceil$ grids in y axis direction. Put one node in each grid until the number of customers meets 50 (If it cannot be evenly divided, leave the grids with the largest x-values in the row with the largest y-value empty). For each customer, we calculate its distance to the depot, and add a noisy $\sim \mathcal{U}(0, 1)$ to the distance. Then we map the distance to $[1, 10]$, round it, and assign it as the demand of the customer. Figure 10 demonstrates problem examples of task G on CVRP and TSP.

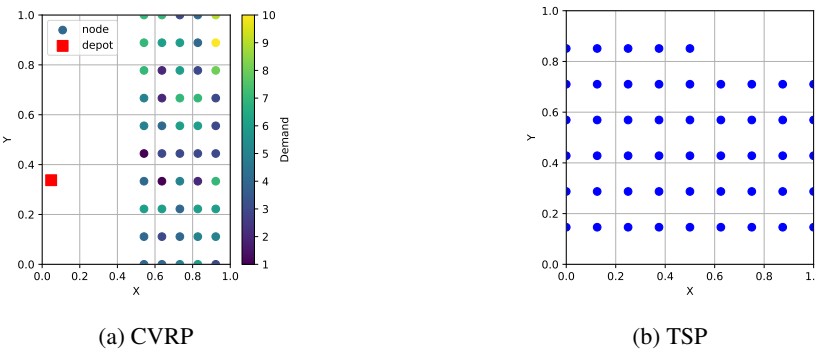

(a) CVRP

(b) TSP

Figure 10: Problem instance sample of task G.

**Task Ring (R):** A ratio $p$ is uniformly sampled from [0.2,0.8]. For each node, we sample an angle $an \, \mathcal{U}(0, 2\pi)$ and a radius $ra = ra_1 + ra_2, ra_1 \sim \mathcal{U}(0.3, 0.4), ra_2 \sim \mathcal{N}(0, 0.05^2)$. The coordinate of the node is set as $(0.5 + ra * cos(an), 0.5 + ra * sin(an))$. If $p \leq 0.5, p \sim \mathcal{U}(0, 1),x \leftarrow x * p$, otherwise $y \leftarrow y * p$. For each customer, we calculate its distance to the depot, and add a noisy $\sim \mathcal{U}(0, 2)$ to the distance. Then we map the distance to $[1, 10]$, round it, and assign it as the demand of the customer. Figure 11 demonstrates problem examples of task R on CVRP and TSP.

## B.2 TASK ORDERS

We randomly shuffle the six tasks five times, getting 5 different task orders.

- Order 1: E $\rightarrow$ C$\rightarrow$ G $\rightarrow$ U $\rightarrow$ R $\rightarrow$ GM.
- Order 2: U $\rightarrow$ GM $\rightarrow$ E $\rightarrow$ R $\rightarrow$ G $\rightarrow$ C.
- Order 3: E $\rightarrow$ G $\rightarrow$ R $\rightarrow$ C $\rightarrow$ U $\rightarrow$ GM.

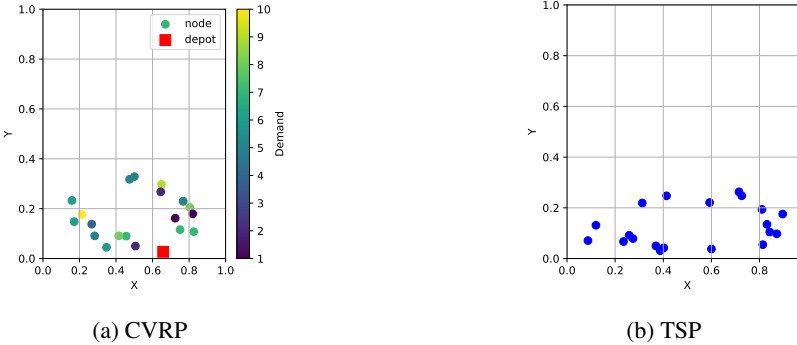

(a) CVRP $\qquad$ (b) TSP

Figure 11: Problem instance sample of task R.

- Order 4: G → GM → E → U → R → C.
- Order 5: G → C → R → U → GM → E.

## C  EXPERIMENT SETTING DETAILS

### C.1  BASELINES

To adapt Li (inter), Li (intra), and Feng to our focused scenarios where the generation of new problem instances is uncontrollable, we maintain one instance buffer for each seen task instead of generating new instances for their experience replay, so that each buffer contains the instances generated during learning the corresponding task. Buffers are updated with reservoir sampling. The buffer size of each task is 200, so that there is a total of 1000 buffered instances that can be used for learning the 6th tasks, aligning with the buffer size of LLR-BC. We set $\alpha = 0.5$ arbitrarily for Li (inter), Li (intra) and Feng, as we did not find a suggested value or method for setting the value of $\alpha$. Feng contains a learnable model parameter $\mathbf{B}$ with fixed size $n \times n$ ($n$ is the problem scale). We did not find the description about the way to handle $\mathbf{B}$ under scenarios where problem scale changes, so we removed $\mathbf{B}$ to make it applicable to our scenarios. To adapt LiBOG in VRP tasks, we directly embed its two key modules, i.e., the intra-task and inter-task consolidation terms, into POMO for lifelong learning. We set the weight of the consolidation term to $10$ for EWC and LiBOG, and the weight of the intra-task consolidation term to $1e − 4$ for LiBOG. For all runs, the solver learns the first task from an identical randomly generated model. As *Restart* learns each task independently under any task order without being affected by the order, we run *Restart* once on each task and calculate metrics under different task orders. Therefore, *Restart*'s APl owns not std. value. As in most cases, POMO can complete the learning on each single task for 200 epochs (Kwon et al., 2020), we let each method train for 200 epochs on each task. To ensure comparable training time across tasks of different problem scales, we vary the number of training problems sampled per epoch: 10,000 for tasks of scale 20, 4,000 for scale 50, and 2,000 for scale 100, with batch sizes of 64, 32, and 16, respectively. Following Kwon et al. (2020), during training, we sample a batch of problem instances and run $N$ parallel solving processes per instance, each corresponding to a different starting node, where $N$ is the problem scale. This results in $N^2$ parallel trajectories per batch, yielding $N^2$ independent states and behaviors per experience batch. The size of the experience batch varies across tasks with different problem scales. We consider this reasonable, as larger-scale tasks are typically more complex and thus require larger experience batches to effectively retain learned knowledge. We experimented with buffering and sampling at the level of individual experiences rather than experience batches. However, when using GPU-based parallelization, operating at the batch level significantly reduces computation time without yielding significant performance differences.

### C.2  EVALUATION METRICS

In designing our evaluation metrics, we follow common practices in lifelong learning (Chaudhry et al., 2018; Wang et al., 2024c). However, when measuring forgetting, we use the test performance obtained immediately after training on a task as the reference, rather than the best performance

before training on the $k$th task. The latter may include effects of backward transfer, which is not the focus of our study. Notably, we report each metric at a given $k$ value rather than averaging across all $k$, as averaging would overemphasize early tasks in forgetting metrics and later tasks in AG, contradicting our assumption that all tasks are equally important.

### C.3 ABLATION STUDY DETAILS

In the ablation version of the confidence measure, negative entropy of action probability distribution, and Top-2 margin, i.e., the absolute difference between top-2 action probability, are used instead of variance. We use the maximal possible entropy to normalize entropy, similar to the normalization of variance. Top-2 margin has a value range of [0,1], so it does not need normalization. All three measurements, including variance, intend to assign higher confidence values (thus lower weights) when the model selects the estimated best action with higher probability, as the consolidation process requires. Results show that, although T2M performs slightly worse in terms of forgetting (arguably because it focuses only on the top two actions and thus discards information about the others), the overall performance of the three measures is very similar. As negative entropy is commonly used as a confidence measurement in recent works, we further calculate the coorelation between varaince and negative entropy. We compute both the variance and the negative entropy for all experiences stored in the buffer after running LLR-BC once under task order 1. The Pearson correlation coefficient between the two measurements is 0.972367 (where 1 indicates perfect positive correlation), showing that, in practice, variance tracks confidence during the lifelong learning process very well. This suggests that, while variance is a simple and effective choice, LLR-BC is not very sensitive to the specific confidence measure. Since LLR-BC is designed to assign higher weights to lower-confidence steps, other confidence measures with similar properties can also be used.

For instance-based buffering, we conduct batch-level buffering, set the buffer size to 20, so that the expected buffered steps are more than the buffer size in the step-based buffer ($20 * \frac{20+20+50+50+100+100}{6} \approx 1133 > 1000$). And 1 (batch of) instances will be sampled from the buffer for each consolidation, i.e., $\mathcal{E} = 1$. With the similar buffer size (in terms of buffered steps (state-behavior pairs) ), instance-based design reduces the instance-diversity of the buffer, potentially leading to inferior performance. The experimental results indicate that our state-based buffer design outperforms the instance-based buffer design, supporting its advantage.

Though reservoir sampling gives each experience an equal probability of being kept in the buffer, since the learning processes of different tasks generate different numbers of experiences, the experiences in the buffer are not uniformly added from all tasks. For tasks with a larger problem scale, more experiences from these tasks are buffered. For one run on task order 1 with buffer size 1000, after learning all tasks, the numbers of experiences in the buffer from each task are: E:162, C:255, G:141, U:78, R:77, GM:287. The ratio of the numbers of buffered experiences is highly correlated with the ratio of the problem scales. We suggest this is reasonable, as tasks with larger scales may be more difficult and need more experiences for consolidating their knowledge and addressing catastrophic forgetting during later learning. After training all tasks with the rescaling version -Res, the numbers of experiences in the buffer from each task are: E:148, C:148, G:144, U:199, R:208, GM:153. Though the buffer is more balanced, it performs (cf. Table 3) very similarly to (or even slightly worse than) the original LLR-BC, suggesting the effectiveness of our design.

### C.4 OTHER DETAILS

Our experiments are conducted on a GPU cluster utilizing a single Nvidia A100 GPU per run. Each lifelong learning method implemented based on POMO[2] (Kwon et al., 2020), Omni[3] (Zhou et al., 2023) , and INViT[4] (Fang et al., 2024) builds upon the official codes released by the original papers.

Based on POMO, each compared method conducts lifelong learning on a single task order once for performance evaluation, requiring approximately one and a half days. For fairness, for each base neural solver, all methods and task orders begin lifelong learning from the same initial model with

---

[2]https://github.com/yd-kwon/POMO
[3]https://github.com/RoyalSkye/Omni-VRP
[4]https://github.com/Kasumigaoka-Utaha/INViT

randomly initialized parameters. Training problem instances are generated on-the-fly at the start of each batch during training.

With 200 training epochs, one task contains $200 * 10000$, $200 * 4000$, $200 * 2000$ problem instances, $\frac{200*10000}{64}$, $\frac{200*4000}{32}$, $\frac{200*2000}{16}$ batchs, and at lest $\frac{200*10000}{64} * 20 = 625000$, $\frac{200*4000}{32} * 50 = 1250000$, $\frac{200*2000}{16} * 100 = 2500000$ experience batchs for scale 20, 50, 100 tasks, respectively. Therefore, the total number of generated experiences after learning the six tasks is 8750000. As $|\mathcal{B}| = 1000$, the buffer contains only $\frac{1000}{8750000} \approx 0.01\%$ of total generated experiences.

## D  DETAILED EXPERIMENT RESULTS

### D.1  MAIN EXPERIMENTS ON CVRP

The detailed metric values in each task order are given in Table 5. Figures 12 and 13 demonstrate the forgetting curve and then learning curve on each current task of each method on task orders 2, 3, 4, and 5. Across all orders, LLR-BC demonstrates the best performance. It aligns with the conclusion in the main text. Additionally, across orders, we found a specific pattern. For *fine-tuning*, learning on GM or C (both scale 100) will lead to significant forgetting of U or R (both scale 20) if they were learned previously, vice versa. It may suggest that the scale difference greatly affects lifelong learning.

The average solution distance and optimality gap (optimal solutions are obtained by HGS) of the solver trained with each method on all tasks follwoing task order 1 are listed in Table 6. Notably, we use the solver trained after the corresponding task to evaluate *Restart*, making it single task method.

### D.2  MAIN EXPERIMENTS ON TSP

The detailed metric values in each task order are given in Table 8. Figures 14 and 15 demonstrate the forgetting curve and then learning curve on each current task of each method on task orders 2, 3, 4, and 5. Across all orders, LLR-BC demonstrates the best performance. It aligns with the conclusion in the main text.

The average solution distance and optimality gap (optimal solutions are obtained by Gurobi) of the solver trained with each method on all tasks follwoing task order 1 are listed in Table 7. Notably, we use the solver trained after the corresponding task to evaluate *Restart*, making it single task method.

### D.3  COMPARISON WITH NON-LIFELONG SETTINGS

Notably, in lifelong learning scenarios, training tasks arise sequentially, and only the current task can be learned. Therefore, multi-task solvers, which learn from training tasks simultaneously and encounter no forgetting issue, are not applicable. To further enhance our experimental study, we implement 2 straightforward multi-task learning methods (batch-level (BL) and epoch-level (EL) task switching) based on POMO, denoted as POMO-MT-BL and POMO-MT-EL, and report the model's performance after learn from all 6 tasks. Results, as follows, demonstrate that the difference is very small, and LLR-BC even outperforms the multi-task settings in the last 4 tasks of CVRP and all tasks of TSP. In addition, we add a comparison with INViT (the recent ancd high-performance, arguable state-of-the-art, cross-distribution and cross-scale neural solver) in the original setting without adaptation to lifelong learning. Specifically, we directly use a model provided by Fang et al. (2024), which is trained with a large budget on the uniform distribution scale 100 task and demonstrates good zero-shot generalization ability to unseen tasks. Comparison results indicate that LLR-BC, with lifelong learning, outperforms INViT that without requiring additional training or lifelong learning mechanisms. It further verifies the importance of lifelong learning and the effectiveness of our proposed LLR-BC. Detailed results are listed in Table 6 and 7.

### D.4  RESULT ON BENCHMARK INSTANCES

We use the identical representative instance set as selected and used by Zhou et al. (2023) Table 9 and 10 present the detailed test performance on CVRPLIB (Set-X) and TSPLIB instances, respectively. Following the same protocol of the main experiments, the reported values are normalized

Table 5: Metric values on each order on CVRP.

| Order | Method | k=3 | | | | | k=6 | | | | |
|---|---|---|---|---|---|---|---|---|---|---|---|
| | | AP | AF | AMF | APl | AG | AP | AF | AMF | APl | AG |
| 1 | Li (inter) | 20.7 | **0.0** | **0.0** | 21.2 | 26.3 | 24.3 | 0.1 | 0.1 | 24.8 | 27.7 |
| | Li (intra) | 33.4 | **0.0** | **0.0** | 39.4 | 47.8 | 33.2 | **0.0** | **0.0** | 37.2 | 41.5 |
| | Feng | 23.2 | 1.4 | 1.4 | 18.8 | 27.6 | 25.7 | 2.2 | 2.9 | 23.9 | 34.4 |
| | Restart | 72.0 | 8.2 | 8.2 | 11.8 | 36.3 | 95.7 | 27.1 | 27.7 | 9.1 | 40.2 |
| | Fine-tuning | 32.8 | 3.2 | 3.2 | 7.5 | 32.8 | 34.5 | 19.9 | 23.1 | 4.6 | 31.3 |
| | EWC | 35.9 | 3.1 | 3.1 | 8.6 | 32.6 | 39.6 | 19.7 | 21.5 | 7.5 | 33.5 |
| | LiBOG | 38.3 | 2.7 | 2.7 | 8.9 | 33.4 | 47.7 | 18.0 | 18.6 | 8.3 | 34.7 |
| | **LLR-BC** | **8.0** | **0.0** | **0.0** | **7.0** | 27.1 | **4.7** | 0.6 | 0.7 | **4.1** | **23.3** |
| 2 | Li (inter) | 28.6 | 0.1 | 0.1 | 30.4 | 47.2 | 33.2 | **0.0** | **0.0** | 34.4 | 42.2 |
| | Li (intra) | 33.1 | 0.5 | 0.5 | 41.1 | 59.9 | 37.4 | **0.0** | 0.2 | 41.8 | 51.6 |
| | Feng | 27.3 | 13.6 | 13.6 | 26.2 | 64.7 | 25.3 | 3.4 | 5.8 | 24.5 | 43.2 |
| | Restart | 44.9 | 92.5 | 92.5 | 8.7 | 60.7 | 42.1 | 37.2 | 64.4 | 9.1 | 52.6 |
| | Fine-tuning | 21.8 | 29.9 | 29.9 | 3.7 | 50.0 | 18.8 | 14.3 | 23.3 | 2.5 | 38.4 |
| | EWC | 19.5 | 33.1 | 33.1 | 4.4 | 48.6 | 19.7 | 12.4 | 23.3 | 4.9 | 37.9 |
| | LiBOG | 22.4 | 29.1 | 29.1 | 4.9 | 51.1 | 22.2 | 12.1 | 21.2 | 5.4 | 39.2 |
| | **LLR-BC** | **2.8** | **0.0** | **0.0** | **3.3** | 37.1 | **2.2** | 0.3 | 0.3 | **1.8** | **22.6** |
| 3 | Li (inter) | 20.2 | **0.0** | **0.0** | 21.3 | 26.8 | 24.0 | **0.0** | **0.0** | 24.8 | 28.2 |
| | Li (intra) | 30.3 | **0.0** | **0.0** | 36.5 | 44.8 | 35.6 | **0.0** | **0.0** | 39.2 | 44.5 |
| | Feng | 19.1 | 2.9 | 2.9 | 15.0 | **21.5** | 25.6 | 1.3 | 2.4 | 24.2 | 26.6 |
| | Restart | 106.8 | 10.7 | 10.7 | 7.8 | 30.9 | 95.7 | 22.8 | 35.7 | 9.1 | 51.9 |
| | Fine-tuning | 35.6 | 4.9 | 4.9 | 6.1 | 24.1 | 33.4 | 21 | 27.4 | 3.9 | 44.4 |
| | EWC | 40.3 | 4.1 | 4.1 | 6.7 | 22.3 | 38.0 | 18.0 | 25.4 | 6.8 | 40.5 |
| | LiBOG | 42.6 | 4.3 | 4.3 | 6.8 | 25.0 | 43.0 | 18.4 | 25.9 | 6.7 | 41.7 |
| | **LLR-BC** | **6.4** | **0.0** | **0.0** | **5.9** | 23.0 | **5.0** | 0.9 | 1.0 | **3.7** | **23.1** |
| 4 | Li (inter) | 36.5 | **0.2** | **0.2** | 40.9 | 62.2 | 40.1 | 0.1 | **0.1** | 42.8 | 52.5 |
| | Li (intra) | 31.7 | 0.3 | 0.3 | 44.8 | 75.9 | 32.5 | **0.0** | **0.1** | 40.3 | 54.8 |
| | Feng | 22.1 | 10.6 | 10.6 | 20.9 | 66.3 | 20.7 | 2.1 | 4.3 | 25.4 | 52.4 |
| | Restart | 35.7 | 44.5 | 44.5 | 10.3 | 77.3 | 42.1 | 26.5 | 35.5 | 9.1 | 52.1 |
| | Fine-tuning | 14.6 | 32.6 | 32.6 | 7.0 | 70.0 | 20.2 | 18.9 | 29.9 | 4.5 | 45.8 |
| | EWC | 19.2 | 28.4 | 28.4 | 8.4 | 69.6 | 27.1 | 18.5 | 23.6 | 7.9 | 46.4 |
| | LiBOG | 19.4 | 29.1 | 29.1 | 8.7 | 69.0 | 26.5 | 20.0 | 24.3 | 7.9 | 46.7 |
| | **LLR-BC** | **7.0** | 0.3 | 0.3 | **6.4** | **58.2** | **4.7** | 0.3 | 0.6 | **4.0** | **32.6** |
| 5 | Li (inter) | 29.4 | **0.1** | **0.1** | 31.7 | 45.1 | 38.6 | **0.0** | **0.0** | 41.2 | 50.1 |
| | Li (intra) | 28.5 | 0.1 | 0.1 | 36.7 | 53.3 | 31.9 | **0.0** | 0.1 | 37.9 | 48.4 |
| | Feng | 26.3 | 1.3 | 1.3 | 24.5 | 47.6 | 25.9 | 7.0 | 7.0 | 23.2 | 38.9 |
| | Restart | 27.2 | 19.3 | 19.3 | 9.5 | 58.6 | 26.7 | 93.1 | 96.5 | 9.1 | 50.5 |
| | Fine-tuning | 11.8 | 11.8 | 11.8 | 7.0 | 50.2 | 10.5 | 25.5 | 36.7 | **3.7** | 50.6 |
| | EWC | 17.8 | 14.5 | 14.5 | 8.2 | 49.1 | 17.0 | 29.0 | 33.7 | 7.5 | 40.9 |
| | LiBOG | 18.3 | 13.5 | 13.5 | 8.6 | 49.0 | 17.3 | 29.9 | 35.7 | 7.7 | 41.7 |
| | **LLR-BC** | **6.5** | 0.4 | 0.4 | **6.7** | **45.3** | **4.2** | 1.6 | 1.6 | **3.7** | **31.8** |

based on the base solution found in our experiments instead of the best-known/optimal solutions reported in existing studies.

Table 6: Avg. solution distance (optimality gap) of each method on CVRP.

| Method | E | C | G | U | R | GM |
|--------|---|---|---|---|---|----|
| Li (inter) | 10.74 (3.5%) | 15.42 (6.1%) | 11.59 (5.2%) | 6.73 (4.1%) | 5.83 (19.1%) | 13.44 (6.6%) |
| Li (intra) | 10.89 (5.0%) | 15.61 (7.3%) | 11.71 (6.4%) | 6.76 (4.5%) | 5.85 (19.4%) | 13.53 (7.3%) |
| Feng | 10.81 (4.2%) | 15.54 (6.9%) | 11.52 (4.5%) | 6.72 (3.9%) | 5.79 (18.1%) | 13.59 (7.7%) |
| Restart | 10.74 (3.5%) | 15.28 (5.1%) | 11.38 (3.3%) | 6.57 (1.6%) | 5.71 (14.4%) | 13.17 (4.3%) |
| Fine-tuning | 10.93 (5.4%) | 15.40 (5.9%) | 11.80 (7.1%) | 7.04 (8.9%) | 5.87 (17.6%) | 13.03 (3.3%) |
| EWC | 10.97 (5.7%) | 15.41 (6.0%) | 11.86 (7.6%) | 7.04 (8.9%) | 5.97 (19.5%) | 13.10 (3.8%) |
| LiBOG | 11.00 (6.0%) | 15.43 (6.1%) | 11.89 (7.9%) | 7.27 (12.3%) | 6.01 (20.3%) | 13.10 (3.8%) |
| INViT | 11.18 (7.8%) | 15.96 (9.8%) | 11.73 (6.5%) | 7.03 (8.6%) | 5.97 (19.7%) | 13.77 (9.2%) |
| POMO-MT-EL | 10.66 (2.7%) | 15.10 (3.9%) | 11.38 (3.3%) | 6.59 (1.8%) | 5.71 (14.5%) | 13.04 (3.3%) |
| POMO-MT-BL | 10.66 (2.8%) | 15.10 (3.9%) | 11.36 (3.1%) | 6.58 (1.7%) | 5.71 (14.5%) | 13.04 (3.4%) |
| LLR-BC | 10.72 (3.3%) | 15.17 (4.3%) | 11.36 (3.1%) | 6.56 (1.5%) | 5.70 (14.1%) | 13.03 (3.3%) |

Table 7: Avg. solution distance (optimality gap) of each method on TSP.

| Method | E | C | G | U | R | GM |
|--------|---|---|---|---|---|----|
| Li (inter) | 5.57 (1.4%) | 6.35 (4.8%) | 5.59 (2.5%) | 3.84 (0.4%) | 1.98 (0.2%) | 4.47 (8.8%) |
| Li (intra) | 5.76 (4.8%) | 6.59 (8.9%) | 5.60 (2.8%) | 3.90 (2.0%) | 2.00 (1.4%) | 4.69 (14.0%) |
| Feng | 5.66 (2.9%) | 6.40 (5.7%) | 5.58 (2.4%) | 3.86 (1.0%) | 1.98 (0.2%) | 4.51 (9.7%) |
| Restart | 5.57 (1.3%) | 6.31 (4.3%) | 5.49 (0.5%) | 3.83 (0.1%) | 1.99 (0.8%) | 4.25 (3.3%) |
| Fine-tuning | 5.67 (3.2%) | 6.32 (4.4%) | 5.72 (4.8%) | 3.88 (1.4%) | 1.98 (0.3%) | 4.20 (2.1%) |
| EWC | 5.67 (3.3%) | 6.36 (5.0%) | 5.61 (2.7%) | 3.89 (1.6%) | 1.98 (0.4%) | 4.23 (2.8%) |
| LiBOG | 5.66 (3.1%) | 6.33 (4.5%) | 5.71 (4.6%) | 3.88 (1.5%) | 1.98 (0.4%) | 4.23 (2.9%) |
| INViT | 5.75 (4.6%) | 6.53 (7.9%) | 5.66 (3.8%) | 3.93 (2.8%) | 2.00 (1.4%) | 4.58 (11.4%) |
| POMO-MT-EL | 5.61 (2.1%) | 6.25 (3.2%) | 5.48 (0.3%) | 3.86 (0.8%) | 1.98 (0.1%) | 4.25 (3.3%) |
| POMO-MT-BL | 5.59 (1.8%) | 6.24 (3.0%) | 5.47 (0.3%) | 3.85 (0.6%) | 1.98 (0.1%) | 4.24 (3.1%) |
| LLR-BC | 5.56 (1.3%) | 6.21 (2.6%) | 5.46 (0.1%) | 3.83 (0.2%) | 1.98 (0.0%) | 4.20 (2.1%) |

Table 8: Metric values on each order on TSP.

| Order | Method | k=3 | | | | | k=6 | | | | |
|---|---|---|---|---|---|---|---|---|---|---|---|
| | | AP | AF | AMF | APl | AG | AP | AF | AMF | APl | AG |
| 1 | Li (inter) | 17.5 | **0.0** | **0.0** | 14.8 | 22.5 | 20.2 | 1.4 | 1.4 | 18.9 | 22.7 |
| | Li (intra) | 42.9 | 1.4 | 1.4 | 51.4 | 59.1 | 45.9 | **0.1** | **0.7** | 50.3 | 54.5 |
| | Feng | 25.3 | 4.8 | 4.8 | 15.9 | 23.5 | 26.5 | 0.7 | 5.5 | 21.6 | 25.1 |
| | Restart | 67.0 | 11.3 | 11.3 | 10.0 | 24.6 | 39.9 | 40.8 | 68 | 7.1 | 57.4 |
| | Fine-tuning | 31.2 | 5.3 | 5.3 | 4.2 | 25.9 | 18.2 | 36.5 | 47.9 | 2.5 | 35.6 |
| | EWC | 26.2 | 4.5 | 4.5 | 5.1 | 27.4 | 17.3 | 22.4 | 22.6 | 4.0 | 29.2 |
| | LiBOG | 30.2 | 3.7 | 3.7 | 5.2 | 28.3 | 19.4 | 15.2 | 22.1 | 4.3 | 38.2 |
| | **LLR-BC** | **2.9** | 0.3 | 0.3 | **2.5** | **19.1** | **1.7** | 0.8 | 0.9 | **1.3** | **21.6** |
| 2 | Li (inter) | 37.6 | **0.0** | **0.0** | 35.8 | 53.7 | 38.0 | **0.5** | 0.8 | 35.8 | 42.6 |
| | Li (intra) | 58.1 | 0.6 | 0.6 | 59.7 | 92.8 | 55.3 | 0.7 | 1.0 | 56.6 | 69.7 |
| | Feng | 29.5 | 1.1 | 1.1 | 25.1 | 56.4 | 26.6 | 2.0 | 2.0 | 22.7 | 35.5 |
| | Restart | 39.3 | 10.0 | 10.0 | 5.4 | 68.7 | 29.7 | 128.4 | 128.4 | 7.1 | 76.0 |
| | Fine-tuning | 16.6 | 7.9 | 7.9 | **2.4** | 61.3 | 13.8 | 42.5 | 42.5 | 2.7 | 59.3 |
| | EWC | 15.4 | 8.5 | 8.5 | 4.2 | 66.1 | 14.5 | 18.9 | 19.2 | 4.7 | 48.1 |
| | LiBOG | 18.3 | 8.0 | 8.0 | 4.0 | 63.0 | 18.7 | 16.8 | 16.8 | 4.7 | 45.1 |
| | **LLR-BC** | **4.5** | 0.1 | 0.1 | 2.8 | **50.6** | **2.5** | **0.5** | **0.6** | **1.6** | **27.6** |
| 3 | Li (inter) | 6.1 | **0.0** | **0.0** | 6.9 | **9.0** | 19.6 | 0.3 | 0.3 | 19.6 | **22.8** |
| | Li (intra) | 22.0 | **0.0** | **0.0** | 29.5 | 27.5 | 46.3 | **0.0** | **0.0** | 51.4 | 57.2 |
| | Feng | 9.4 | 3.7 | 3.7 | **1.3** | 11.3 | 25.6 | 3.9 | 4.3 | 21.5 | 23.8 |
| | Restart | 49.9 | 56.8 | 56.8 | 3.1 | 70.5 | 39.9 | 14.6 | 42.8 | 7.1 | 72.9 |
| | Fine-tuning | 19.9 | 29.2 | 29.2 | 1.8 | 17.6 | 15.4 | 15.4 | 30.5 | 2.2 | 41.5 |
| | EWC | 19.8 | 19.9 | 19.9 | 2.6 | 16.7 | 17.0 | 10.8 | 19.0 | 3.8 | 37.4 |
| | LiBOG | 20.6 | 15.6 | 15.6 | 2.1 | 16.7 | 17.4 | 11.8 | 20.2 | 4.3 | 37.2 |
| | **LLR-BC** | **2.2** | **0.0** | **0.0** | 1.9 | 10.0 | **3.0** | 0.6 | 0.9 | **1.8** | 22.9 |
| 4 | Li (inter) | 98.5 | **0.0** | **0.0** | 104.5 | 158.1 | 109.6 | **0.0** | **0.0** | 119.1 | 146.4 |
| | Li (intra) | 73.9 | 0.4 | 0.4 | 100.2 | 203.6 | 63.7 | **0.0** | 0.1 | 79.1 | 122.6 |
| | Feng | 20.6 | **0.0** | **0.0** | 21.1 | 186.7 | 17.9 | **0.0** | 1.2 | 18.5 | 84.6 |
| | Restart | 40.9 | 46.0 | 46.0 | **7.0** | 156.2 | 29.7 | 37.6 | 47.5 | 7.1 | 85.7 |
| | Fine-tuning | 25.1 | 32.2 | 32.2 | 7.4 | 164.3 | 15.5 | 33.2 | 33.2 | 4.4 | 85.2 |
| | EWC | 34.1 | 34.1 | 34.1 | 8.6 | 160.8 | 22.8 | 23.2 | 31.2 | 7.0 | 77.6 |
| | LiBOG | 32.4 | 34.4 | 34.4 | 8.7 | **154.5** | 22.2 | 25.5 | 33.1 | 7.5 | 74.5 |
| | **LLR-BC** | **8.9** | 1.1 | 1.1 | 7.7 | 154.6 | **5.0** | 1.4 | 1.4 | **4.2** | **68.9** |
| 5 | Li (inter) | 75.1 | **0.1** | **0.1** | 100.5 | 161.7 | 95.2 | **0.0** | **0.1** | 115.0 | 153.6 |
| | Li (intra) | 42.6 | 2.0 | 2.0 | 69.8 | 127.3 | 60.0 | 0.2 | 0.8 | 76.6 | 107.9 |
| | Feng | 17.3 | 1.5 | 1.5 | 13.1 | 100.0 | 23.9 | 2.6 | 5.8 | 22.8 | 58.5 |
| | Restart | 14.0 | 11.1 | 11.1 | 8.8 | 99.5 | 19.5 | 30.9 | 41.5 | 7.1 | 69.9 |
| | Fine-tuning | 11.5 | 9.1 | 9.1 | 9.4 | 99.1 | 11.3 | 16.7 | 28.9 | 5.7 | 65.4 |
| | EWC | 21.5 | 12.0 | 12.0 | 8.9 | 102.5 | 20.0 | 17.7 | 31.0 | 8.1 | 68.3 |
| | LiBOG | 18.7 | 9.4 | 9.4 | **8.6** | **92.1** | 18.2 | 16.9 | 22.0 | 8.2 | **63.5** |
| | **LLR-BC** | **7.5** | 0.8 | 0.8 | 9.1 | 106.3 | **4.8** | 0.5 | 1.7 | **5.3** | 64.5 |

Table 9: Test performance on CVRPLIB instances.

| Instance | Li (intra) | Li (inter) | Feng | Restart | Fine-tuning | EWC | LiBOG | LLR-BC |
|---|---|---|---|---|---|---|---|---|
| X-n101-k25 | 38.33 | 19.09 | 42.64 | 8.16 | 17.92 | 0 | 10.56 | 8.93 |
| X-n153-k22 | 52.41 | 12.65 | 30.32 | 38.03 | 0 | 21.71 | 43.18 | 23.72 |
| X-n200-k36 | 26.28 | 21.48 | 10.85 | 19.54 | 8.13 | 7.02 | 13.22 | 0 |
| X-n251-k28 | 12.71 | 12.98 | 14.37 | 14.26 | 0 | 14.25 | 9.23 | 1.47 |
| X-n303-k21 | 83.59 | 44.96 | 18.77 | 66.62 | 10.03 | 38.12 | 27.47 | 0 |
| X-n351-k40 | 43.88 | 65.26 | 0 | 118.54 | 5.53 | 6.5 | 18.25 | 0 |
| X-n401-k29 | 29.12 | 39.31 | 30.61 | 23.33 | 1.48 | 16.77 | 21.85 | 0 |
| X-n459-k26 | 81.01 | 60.34 | 66.94 | 76.71 | 0 | 9.8 | 8.78 | 20.85 |
| X-n502-k39 | 32.57 | 87.95 | 31.46 | 9.86 | 0 | 4.42 | 21.88 | 20.08 |
| X-n548-k50 | 25.36 | 6.86 | 5.78 | 35.99 | 0 | 9.31 | 6.18 | 4.33 |
| X-n599-k92 | 55.1 | 25.09 | 6.62 | 27.25 | 17.92 | 7.76 | 25.96 | 0 |
| X-n655-k131 | 66.91 | 60.46 | 15.64 | 31.02 | 26.97 | 0 | 11.66 | 0.77 |
| X-n701-k44 | 34.08 | 14.51 | 21.06 | 58.96 | 0 | 30.82 | 9.88 | 13.01 |
| X-n749-k98 | 57.64 | 46.36 | 41.03 | 43.41 | 20.44 | 0 | 13.76 | 7.94 |
| X-n801-k40 | 17.03 | 24.97 | 20.79 | 29.82 | 0 | 2.3 | 1.55 | 14.34 |
| X-n856-k95 | 21.53 | 34.56 | 47.96 | 50.49 | 29.5 | 13.87 | 35.03 | 0 |
| X-n895-k37 | 119.88 | 53.02 | 17.84 | 71.45 | 0 | 33.18 | 26.84 | 21 |
| X-n957-k87 | 57.95 | 74.26 | 61.91 | 58.04 | 24.37 | 9.06 | 27.86 | 0 |
| X-n1001-k43 | 34.88 | 43.83 | 35.69 | 48.51 | 0 | 0.5 | 9.78 | 13.28 |
| Mean (Std.) | 46.86(26.95) | 39.37(23.18) | 27.38(18.53) | 43.68 (27.37) | 8.54 (10.71) | 11.86 (11.59) | 18.05 (10.86) | **7.88 (8.75)** |

Table 10: Test performance on TSPLIB instances.

| Instance | Li (intra) | Li (inter) | Feng | Restart | Fine-tuning | EWC | LiBOG | LLR-BC |
|---|---|---|---|---|---|---|---|---|
| kroA100 | 115.35 | 56.74 | 44.05 | 86.88 | 40.56 | 42.67 | 75.6 | 0 |
| kroA150 | 85.3 | 56.6 | 11.67 | 29.43 | 28.06 | 0 | 7.58 | 12.15 |
| kroA200 | 116.89 | 73.23 | 96.39 | 101.65 | 0 | 16.93 | 44.99 | 48.77 |
| kroB200 | 90.12 | 46.27 | 55.42 | 105.78 | 10.86 | 34.53 | 12.47 | 0 |
| ts225 | 145.3 | 27.3 | 102.23 | 152.45 | 60.52 | 70.33 | 24.24 | 0 |
| tsp225 | 76.08 | 31.3 | 28.75 | 97.43 | 9.61 | 0 | 15 | 2.47 |
| pr226 | 38.69 | 30.16 | 42.36 | 17.84 | 0 | 5.95 | 16.59 | 16.12 |
| pr264 | 279.81 | 164.86 | 164.17 | 47.94 | 36.59 | 0 | 34.88 | 46.66 |
| a280 | 119.41 | 43.49 | 80.83 | 72.55 | 8.98 | 0 | 28.86 | 19.19 |
| pr299 | 127.96 | 64.42 | 65.24 | 79.27 | 0 | 5.09 | 6.53 | 16.29 |
| lin318 | 127.91 | 61.94 | 104.45 | 97.15 | 0 | 16 | 46.94 | 32.34 |
| rd400 | 128.69 | 53.32 | 55.73 | 81.68 | 18.25 | 0 | 25.14 | 20.14 |
| fl417 | 28.2 | 58.18 | 80.35 | 40.69 | 3.67 | 30.88 | 0 | 21.12 |
| pr439 | 120.37 | 50.77 | 61.96 | 126.81 | 29.1 | 3.8 | 0 | 0.14 |
| pcb442 | 157.56 | 79.02 | 81.29 | 108.27 | 0 | 1.17 | 17.31 | 11.12 |
| d493 | 40 | 52.33 | 89.3 | 253.56 | 443.1 | 280.94 | 123.82 | 0 |
| u574 | 127.96 | 43.75 | 73.05 | 92.14 | 19.11 | 3.25 | 10.8 | 0 |
| rat575 | 152.37 | 51.56 | 77.56 | 122.04 | 0 | 24.43 | 20.29 | 52.95 |
| p654 | 150.62 | 56.15 | 60.28 | 45.81 | 0 | 11.12 | 5.04 | 26.84 |
| d657 | 126.42 | 50.54 | 104.57 | 267.62 | 136.78 | 81.08 | 101.14 | 0 |
| u724 | 161.09 | 58.65 | 83.95 | 66.06 | 30.1 | 0 | 2.83 | 25.28 |
| rat783 | 155.97 | 65.08 | 75.42 | 135.86 | 0 | 10.48 | 23.03 | 34.46 |
| pr1002 | 147.73 | 68.96 | 89.49 | 65.53 | 2.33 | 0.67 | 0 | 29.88 |
| Mean (Std. ) | 122.60(51.78) | 58.46(26.55) | 75.15(30.83) | 99.76(61.06) | 38.16(93.36) | 27.80(59.58) | 27.96(32.29) | **18.08(16.98)** |

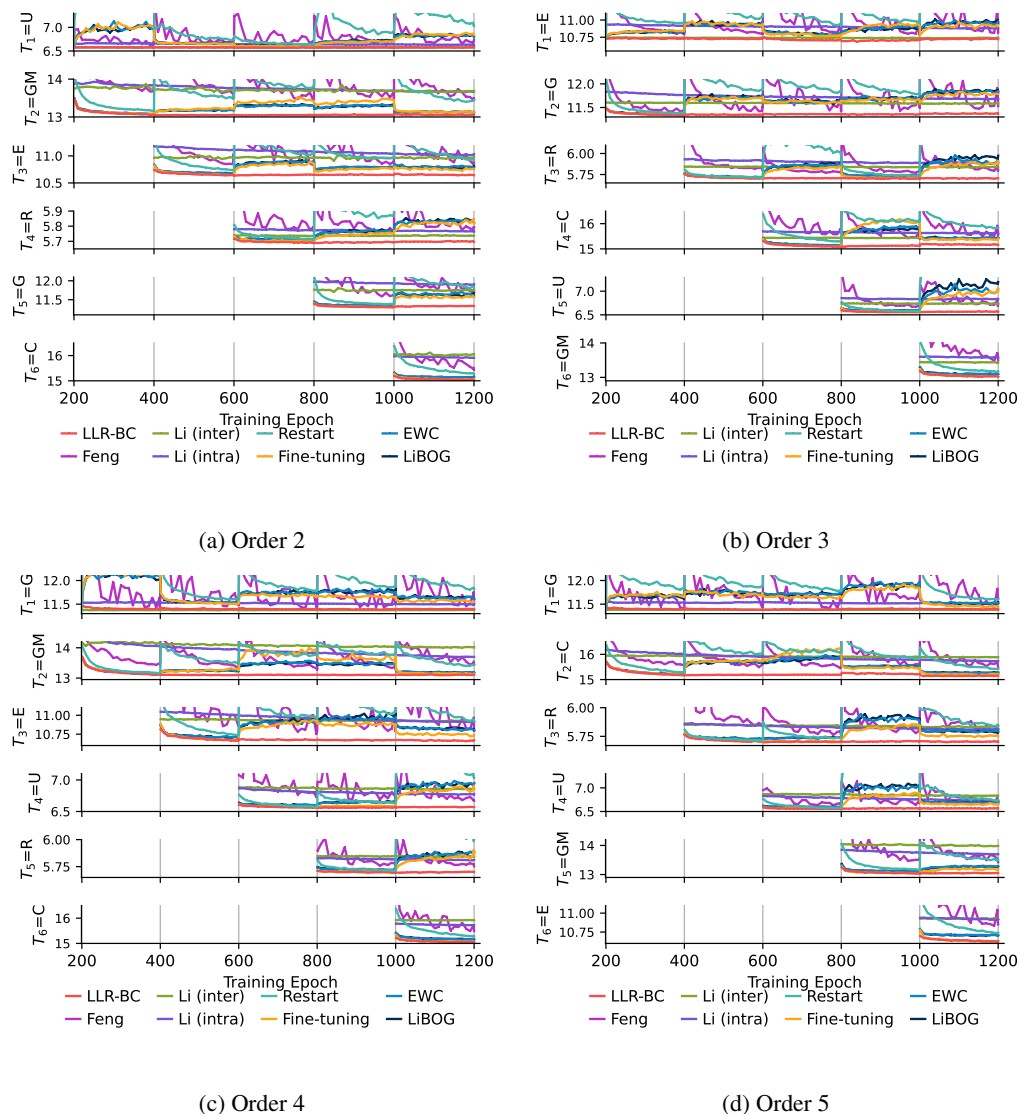

(a) Order 2            (b) Order 3

(c) Order 4            (d) Order 5

Figure 12: Forgetting curve on CVRP task orders 2-5, measured by average solution distance (vertical axis). Epochs 0–200 (first task) are omitted as no forgetting occurs. *Restart* is excluded due to significantly higher forgetting than other methods.

### D.5 HYPERPARAMETER SENSITIVITY ANALYSIS DETAILS

As Table 3 demonstrated, on CVRP, increasing $\alpha$ reduces AF and AMF while increasing APl as expected, as it leads to greater emphasis on stability over plasticity, which aligns with expectations since a higher weight in the consolidation loss term will lead to more focus on stability rather than plasticity. In contrast, on the TSP, increasing $\alpha$ results in worse performance across evaluation metrics. A potential reason is that TSP tasks are more similar to each other than those in CVRP, as TSP does not involve differences in demand distribution. This higher similarity increases the likelihood of beneficial backward transfer when learning a new task. In such cases, using a lower consolidation weight could be more advantageous. Nevertheless, LLR-BC consistently outperforms all baseline methods under each tested setting (cf. Table 1). No significant pattern in performance changing is found by varying $|\mathcal{E}|$ or $|\mathcal{B}|$. But the impact of changing them is substantially small, verifying the robustness of LLR-BC.

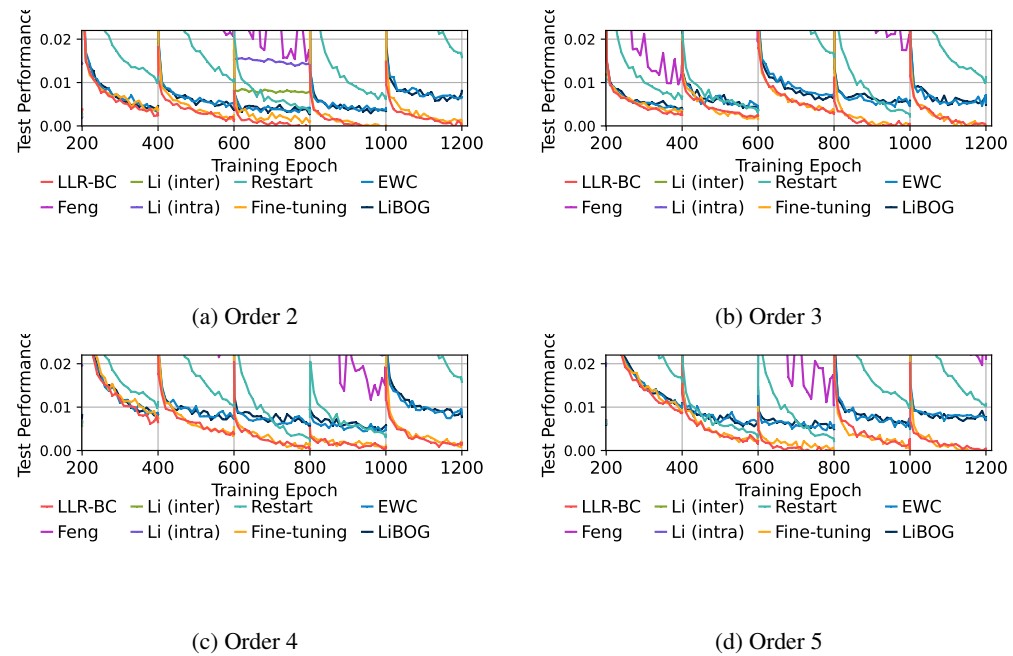

(a) Order 2

(b) Order 3

(c) Order 4

(d) Order 5

Figure 13: Test performance on the current task during lifelong learning on CVRP task orders 2-5.

## D.6 DETAILS OF APPLICABILITY EXPERIMENTS

We evaluate LLR-BC on Omni (Zhou et al., 2023) and INViT Fang et al. (2024) to verify its applicability (cf. Section 4.6). Omni is designed to produce a strong initial model that can quickly adapt to new tasks, through meta-learning. Accordingly, we adopt the meta-learned initial model provided by Zhou et al. (2023) as the starting model for the first task, and run lifelong learning with 10 epochs for each task, following their protocol. INViT aims at learning from one task to achieve strong zero-shot generalization to other tasks. For INViT, we sequentially train on tasks for 100 epochs each. As the default setting (Fang et al., 2024), data augmentation with size 8 is used in INViT. Hyperparameters of LLR-BC are set as follows: $|\mathcal{B}| = 1000$, $|\mathcal{E}| = 16$, and $\alpha = 100$, identical to the one used on POMO. Hyperparameters of base neural solvers are set identically to the original papers. All other settings are consistent with those used with POMO, as above. Since our goal is not to compare performance differences across base neural solvers, we apply independent normalization for each base neural solver, i.e., for different base neural solvers the value of $d_j^*$ is different. This allows us to demonstrate the applicability of LLR-BC across different base solvers more clearly, without the distraction from performance differences in base neural solvers.

For Omni, only on APl of CVRP, LLR-BC is slightly higher than *fine-tuning*. It is considered reasonable as Omni is designed to quickly adapt and owns outstanding plasticity, which overwrites the benefit from the potential forward transferring of LLR-BC. For INViT, LLR-BC outperforms *fine-tuning* on most metrics, except for the forgetting measures on CVRP. A potential reason for this weaker performance is that TSP tasks are highly similar to one another (relative to the greater diversity among CVRP tasks), and INViT is known to exhibit strong ability to learn general knowledge of relatively similar tasks from one task (Fang et al., 2024). In such cases, incorporating prior experience that is not directly relevant to the current learning task may introduce distracting signals, which can outweigh the benefits in mitigating forgetting.

Notably, the performance of LLR-BC can be further improved by tuning its hyperparameters for each individual base neural solver. Overall, LLR-BC outperforms *fine-tuning* across most evaluation metrics, regardless of the used base neural solver, thereby confirming its broad applicability.

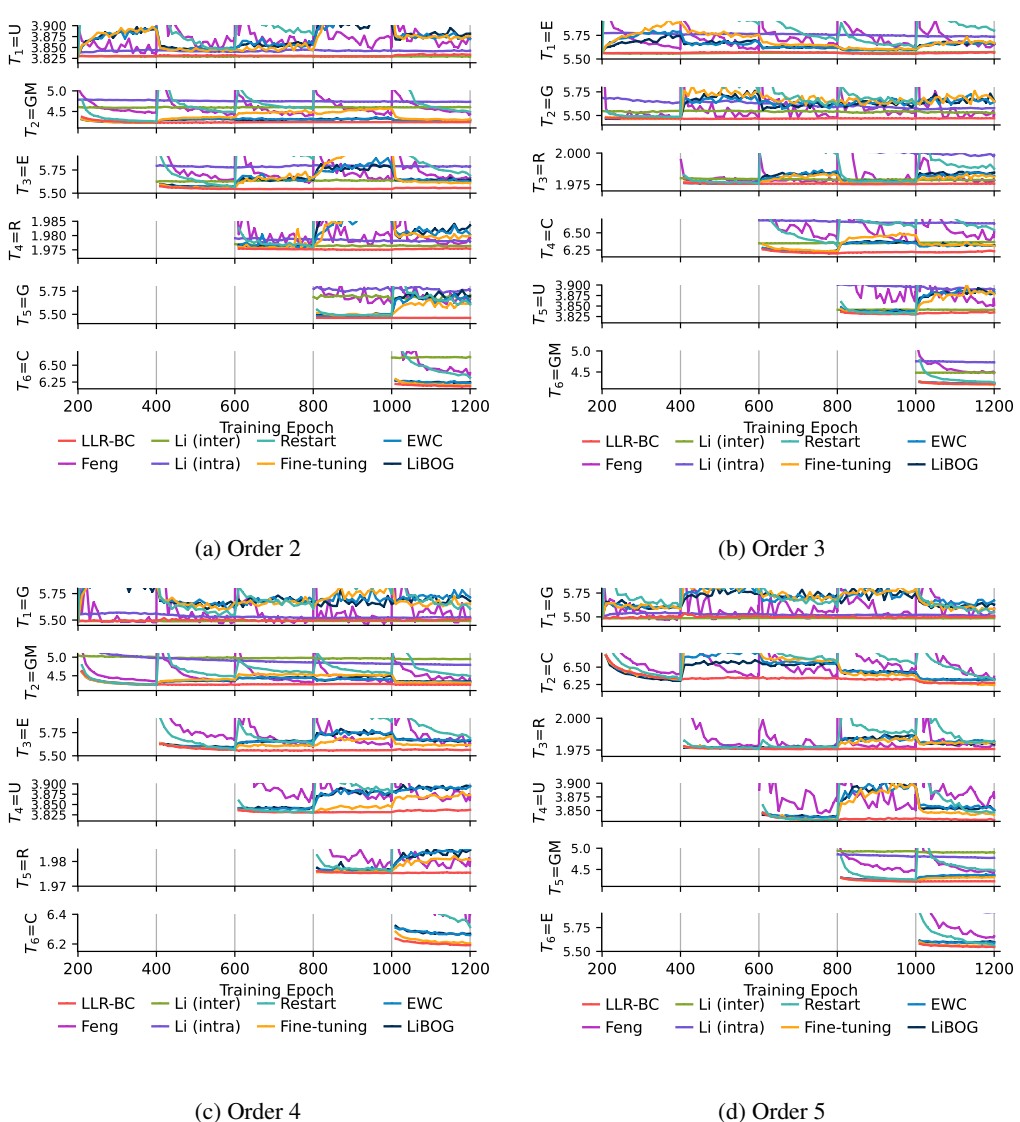

(a) Order 2

(b) Order 3

(c) Order 4

(d) Order 5

Figure 14: Forgetting curve on TSP task orders 2-5, measured by average solution distance (vertical axis). Epochs 0–200 (first task) are omitted as no forgetting occurs. *Restart* is excluded due to significantly higher forgetting than other methods.

# E  LLM USAGE STATEMENT

We used ChatGPT (GPT-5) only as an assistive tool for grammar checking and language polishing. The model was not involved in research ideation, algorithm design, experiment execution, or result analysis. All scientific content and conclusions are entirely the work of the authors.

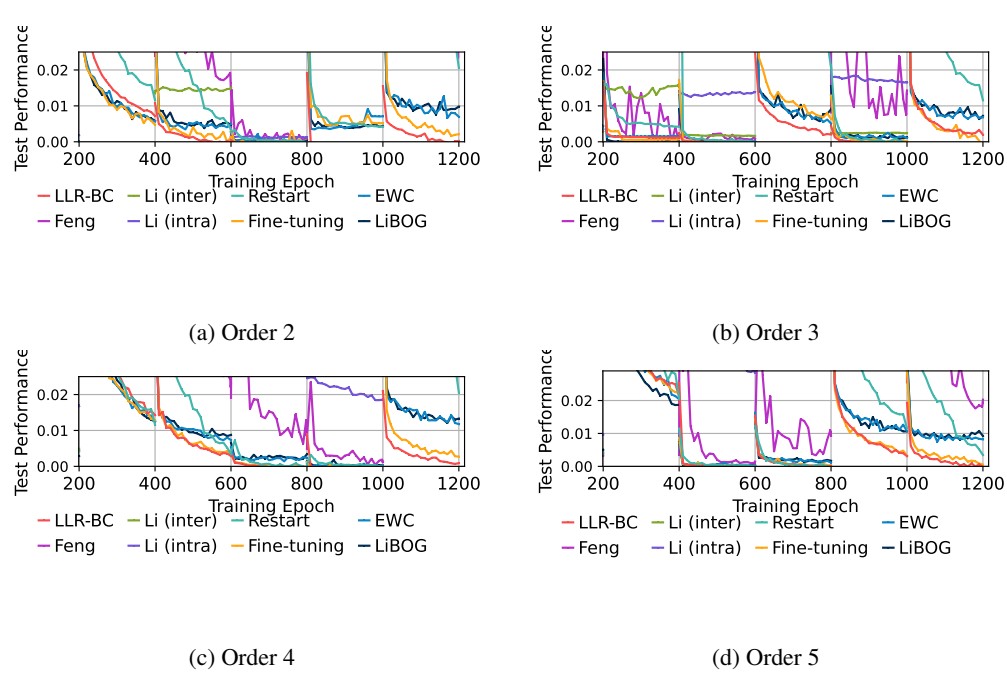

Figure 15: Test performance on the current task during lifelong learning on TSP task orders 2-5.

