# OpenReview forum: "Lifelong Learning with Behavior Consolidation for Vehicle Routing"
_ICLR.cc/2026/Conference — ICLR 2026 Poster_

### Official Review · Reviewer_iZkt · 2025-10-20

**Soundness:** 3
**Presentation:** 3
**Contribution:** 2
**Rating:** 4
**Confidence:** 4

**Summary:**

This paper introduces a lifelong learning setting for the VRP domain and proposes the Lifelong Learning Router with Behavior Consolidation (LLR-BC) to address the issue of catastrophic forgetting. Specifically, LLR-BC incorporates Confidence-aware Experience Weighting (CaEW) to prioritize critical experiences and Decision-seeking Behavior Consolidation (DsBC) to emphasize replicating past decisions. Extensive experiments on TSP and CVRP benchmarks demonstrate the effectiveness of LLR-BC.

**Strengths:**

* The paper is overall well-written and easy to follow.
* The studied topic, i.e., generalization of neural VRP solvers, is important, and the proposed setting is sound.
* The experiments are generally extensive and solid, demonstrating that the catastrophic forgetting issue is significantly mitigated, as shown in Fig. 3.

**Weaknesses:**

* This paper applies lifelong learning to the VRP domain, incorporating a reweighting mechanism and reverse KL divergence. The overall technical novelty appears somewhat limited.
* In line with the no free lunch principle, while the proposed method mitigates catastrophic forgetting on previous tasks, it may incur trade-offs elsewhere. Could the authors clarify what is sacrificed or traded off in their approach or setting?
* The caption of Fig. 2 should explicitly explain all notations.
* The writing in two parts could be improved:
  * In Section 3.1, the authors should first introduce the principle of reservoir sampling.
  * In Section 3.3, the connection between KLD and RKLD should be highlighted as a change in direction from $KL(\pi_\theta | \mathcal{P}) \to KL(\mathcal{P} | \pi_\theta)$. The authors should also mathematically explain why this directional change fundamentally alters model behaviour during learning.
* The experiments primarily compare lifelong learning baselines, but lack a comprehensive evaluation against traditional and neural VRP solvers. Could the authors include direct comparisons with recent generalizable neural VRP solvers, e.g., by multi-task training them on the same task distribution?
* Minor on line 152: (i.e., )

----

I intend to assign a borderline rejection in the first round of evaluation and would like to see the authors’ rebuttal addressing my main concerns, particularly the experimental comparisons.

**Questions:**

* Is Fig. 1 a conceptual illustration or based on real experimental results?
* Could LLR-BC be applied to SL-based solvers as well?
* The experience buffer is state-based, why not consider an instance-based design?
* Is there any empirical evidence supporting the claims made in lines 236–240?

---

> ### Author Response · Authors · 2025-11-21
> **Response to Reviewer iZkt**
>
> We are grateful to the reviewers for their constructive feedback, which has greatly helped us improve the clarity and quality of this work. We have revised the paper accordingly and highlighted all changes in blue. Below, we provide point-by-point responses to all comments.
>
> **W1: This paper applies lifelong learning to the VRP domain, incorporating a reweighting mechanism and reverse KL divergence. The overall technical novelty appears somewhat limited.**
>
> Compared with existing studies of lifelong learning in neural VRP solvers [1,2], our technical novelties in method design are as follows:
> - **Formulation of experience.** We treat each step (a ⟨state, behavior⟩ pair) in solving an instance as one experience, whereas [1,2] treat the entire instance as one experience. This step-based formulation provides finer granularity and is more effective under the same computational budget. We conducted an additional ablation study (Section 4.4, Table 3) comparing LLR-BC with an instance-based variant (LLR-BC-IB), and the results confirm the superior effectiveness of our step-based design.
> - **Additional cost for generating experience.** Our method reuses previously buffered experiences for consolidation instead of regenerating new ones as in [1,2]. This reduces computational cost and also makes our approach applicable to scenarios where instance generation is uncontrollable, in which **the methods of [1,2] cannot be directly applied**.
> - **How consolidation is performed.** To preserve previously learned knowledge when learning a new task, [2] constrains the parameters of the current model to stay close to those of the previous model, in a way similar to EWC. As shown in experiments of our original paper (cf. Table 1), parameter-level regularization (EWC, LiBOG) leads to inferior performance. The method in [1] encourages the current and previous models to behave similarly **on the new task**. In contrast, our method encourages the current and previous models to exhibit similar behavior **on the previous tasks**. Behaviors on previous tasks encode more direct knowledge about those tasks and therefore provide more effective signals for consolidation. Our method thus achieves better performance than [1] and [2] in the following reported experiments.
> - **Additional design components.** In addition, we introduce a new confidence-based weighting scheme and an new RKLD-based decision-seeking consolidation objective, as described in the original manuscript (Sections 3.2 and 3.3). The new components further improve the performance of our method (cf. Section 4.4 and Table 3 of the original paper).
>
> Comparison with [3,4], as well as the ablation studies on experience weighting and RKLD, are already included in our original paper (cf. Tables 1 and 3). **In addition, we have adapted and implemented the methods in [1] and [2] and compared them with our approach to further verify the effectiveness of our design.** The results, shown below, demonstrate that generally our method achieves the best performance. Notably, although the methods of [1] exhibit less forgetting (AF, AMF), their plasticity (APl) for learning the new task is poor, because half of the epochs on the new task are spent learning purely from instances of previous tasks. As a result, their overall performance on seen (AP) and unseen (AG) tasks is inferior. We have added these new experimental results to the paper and revised Section 4.4, Tables 1 and 3, and Appendix C.3 accordingly.
>
> **CVRP (mean (std.) over 5 task orders):**
> |Method|AP|AF|AMF|APl|AG|
> |-|-|-|-|-|-|
> Li (inter) [1]|32.0 (6.8)|**0.0** (**0.0**)|**0.0** (**0.0**)|33.6 (7.7)|40.1 (10.5)|
> Li (intra) [1]|34.1 (2.1)|**0.0** (**0.0**)|0.1 (0.1)|39.3 (1.7)|48.2 (4.8)
> Feng [2]|24.6 (2.0)|3.2 (2.0)|4.5 (1.7)|24.2 (0.7)|39.1 (8.6)
> **LLR-BC**|**4.2** (**1.0**)|0.7 (0.5)|0.8 (0.4)|**3.5** (**0.8**)|**26.7** (**4.5**)
>
> **TSP (mean (std.) over 5 task orders):**
> |Method|AP|AF|AMF|APl|AG|
> |-|-|-|-|-|-|
> Li (inter) [1]|56.5 (38.3)|0.4 (0.5)| **0.5** (0.5)|61.7 (45.6)|77.6 (59.6)
> Li (intra) [1]|54.2 (7.2)|**0.2** (**0.3**)|**0.5** (**0.4**)|62.8 (12.5)|82.4 (27.7)
> Feng [2]|24.1 (3.2)|1.8 (1.4)|3.8 (1.9)|21.4 (**1.6**)|45.5 (23.2)
> **LLR-BC**|**3.4** (**1.3**)|0.8 (**0.3**)|1.1 (**0.4**)|**2.8** (**1.6**)|**41.1** (**21.0**)
>
>
>
> [1] Jingwen Li, et al. Enhancing the cross-size generalization for solving vehicle routing problems via continual learning, 2024. URL https://openreview.net/forum?id=WdvT2UgsTK.
>
> [2] Shaodi Feng, et al. Lifelong learner: Discovering versatile neural solvers for vehicle routing problems, 2025. URL https://arxiv.org/abs/2508.11679.
>
> [3] James Kirkpatrick, et al. Overcoming catastrophic forgetting
> in neural networks. Proceedings of the National Academy of Sciences, 114(13):3521–3526, 2017.
>
> [4] Jiyuan Pei, et al. Libog: Lifelong learning for black-box optimizer
> generation, 2025. URL https://arxiv.org/abs/2505.13025.

---

> > ### Author Response · Authors · 2025-11-21
> > **Response to Reviewer iZkt cnt.**
> >
> > **W2: In line with the no free lunch principle, while the proposed method mitigates catastrophic forgetting on previous tasks, it may incur trade-offs elsewhere. Could the authors clarify what is sacrificed or traded off in their approach or setting?**
> >
> > We agree that mitigating catastrophic forgetting leads to trade-offs in other aspects. Although this effect is not significant in our experiments, the following aspects may be sacrificed:
> > 1. **Computational resource cost:** LLR-BC requires more storage for the experience buffer and more computational time, mainly for obtaining the model’s new behaviors on old states, as discussed in the paper.
> > 2. **Plasticity in cases where tasks are highly different:** The good plasticity demonstrated by LLR-BC in our experiments lies in knowledge transfer between tasks, i.e., knowledge learned from previous tasks improves learning on new tasks. However, there may be cases where tasks are highly different, so that previous knowledge becomes misleading. In those cases, the plasticity of LLR-BC could be sacrificed to some degree.
> >
> > **W3: The caption of Fig. 2 should explicitly explain all notations.**
> >
> > We sincerely appreciate the reviewer’s comment, which helps improve the readability of our paper. We have added the corresponding notations to the caption of Fig. 2.
> >
> > **W4: The writing in two parts could be improved: In Section 3.1, the authors should first introduce the principle of reservoir sampling. In Section 3.3, the connection between KLD and RKLD should be highlighted as a change in direction from. The authors should also mathematically explain why this directional change fundamentally alters model behaviour during learning.**
> >
> > Thank you for the comment on improving the writing. We have revised the corresponding parts of our paper accordingly.
> >
> >
> > **W5: The experiments primarily compare lifelong learning baselines, but lack a comprehensive evaluation against traditional and neural VRP solvers. Could the authors include direct comparisons with recent generalizable neural VRP solvers, e.g., by multi-task training them on the same task distribution?**
> >
> > In this revision, we have added additional comparisons with
> > 1. **INViT [5]** (a recent high-performance cross-distribution and cross-scale neural solver) in its original setting without adaptation to lifelong learning. We evaluate it using a model provided by [5], which is trained with a large budget on a uniform-distribution, scale-100 task and demonstrates strong zero-shot generalization ability to unseen tasks.
> > 2. **Two straightforward multi-tasking settings of POMO:**
> >    1. Training by switching to a random one of the six tasks at the beginning of each epoch, denoted **POMO-MT-EL**.
> >    2. Training by switching to a random one of the six tasks at the beginning of each batch, denoted **POMO-MT-BL**.
> >
> > LLR-BC outperforms INViT, indicating that lifelong learning on new tasks provides a significant advantage over existing generalizable neural solvers. LLR-BC also performs very similarly to, and even better on four CVRP tasks and all TSP tasks than, the multi-tasking settings of POMO. This further verifies the effectiveness of LLR-BC. We have added the new experimental results as Tables 6 and 7 in the Appendix of our revised paper.
> >
> >
> >
> >
> > **Avg. solution distance on test set of each task of CVRP.**
> > |Method|E|C|G|U|R|GM
> > |-|-|-|-|-|-|-
> > POMO-MT-EL|**10.6567**|**15.1016**|11.3768|6.5851|5.7132|13.0378
> > POMO-MT-BL|10.6618|15.1032|11.3594|6.5784|5.7140|13.0448
> > INViT|11.1823|15.9606|11.7299|7.0265|5.9726|13.7742
> > **LLR-BC**|10.7198|15.1676|**11.3577**|**6.5640**|**5.6964**|**13.0321**
> >
> > **Avg. solution distance on test set of each task of TSP.**
> > |Method|E|C|G|U|R|GM
> > |-|-|-|-|-|-|-
> > POMO-MT-EL|5.6101|6.2478|5.4750|3.8550|1.9765|4.2491
> > POMO-MT-BL|5.5924|6.2359|5.4735|3.8486|1.9762|4.2412
> > INViT|5.7489|6.5308|5.6635|3.9311|2.0030|4.5812
> > **LLR-BC**|**5.5566**|**6.2116**|**5.4639**|**3.8328**|**1.9755**|**4.1945**
> >
> > Furthermore, though working under different scenario assumptions, we acknowledge that studies of multi-task neural VRP solvers, e.g., [6,7], are very close to our topic and can provide insights for our future work on lifelong learning of neural VRP solvers.
> >
> > [5] Han Fang, Zhihao Song, Paul Weng, and Yutong Ban. INViT: A generalizable routing problem solver with invariant nested view transformer. In Proceedings of the 41st International Conference on Machine Learning, ICML’24. JMLR, 2024.
> >
> > [6] Fei Liu, Xi Lin, Zhenkun Wang, Qingfu Zhang, Tong Xialiang and Mingxuan Yuan. Multi-task learning for routing problem with cross-problem zero-shot generalization. In Proceedings of the 30th ACM SIGKDD Conference on Knowledge Discovery and Data Mining, pp. 1898-1908.
> >
> > [7] Suyu Liu, Zhiguang Cao, Shanshan Feng and Yew-Soon Ong. A Mixed-Curvature based Pre-training Paradigm for Multi-Task Vehicle
> > Routing Solver. In Proceedings of the 41st International Conference on Machine Learning, ICML’25. JMLR, 2025.

---

> ### Author Response · Authors · 2025-11-21
> **Response to Reviewer iZkt cnt.**
>
> **W6: Minor on line 152: (i.e., )**
>
> We appreciate the reviewer’s careful reading and have revised the sentence accordingly.
>
> **Q1: Is Fig. 1 a conceptual illustration or based on real experimental results?**
>
> Fig. 1 is a conceptual illustration. We did not use real experimental results because (i) there are significant differences in distance scales between tasks, and (ii) fluctuations during training would hinder the clarity of the illustration.
>
>
> **Q2: Could LLR-BC be applied to SL-based solvers as well?**
>
> Yes, it can be applied to SL-based solvers with only very small adaptations:
> 1. In SL there is no explicit definition of “state”. Therefore, an experience may no longer be a ⟨state, behavior⟩ pair as in RL. In general, an experience can be any ⟨model input, model output⟩ pair. The detailed definition depends on the specific design of the solver.
> 2. Depending on how the model output is defined, the decision-confidence measure may need to be adapted.
>
>
> **Q3: The experience buffer is state-based, why not consider an instance-based design?**
>
> We thank the reviewer for raising this thoughtful point. With the same buffer size (in terms of buffered steps (<state, behavior> pairs) ), instance-based design reduces the instance-diversity of the buffer, potentially leading to inferior performance. We have added an ablation variant in our ablation study (cf. Section 4.4, Table 3, and Appendix C.3) that conducts instance-based experience buffering, denoted LLR-BC-IB. Specifically, LLR-BC-IB treats all steps (⟨state, behavior⟩ pairs) for solving a batch of instances (since the buffer is at the batch level) as one experience. We set the buffer size to 20 so that the expected number of buffered steps is larger than in LLR-BC (20 * $\frac{20+20+50+50+100+100}{6}$ ≈ 1133 > 1000 ). The experimental results, shown below, indicate that our state-based buffer design outperforms the instance-based buffer design, verifying the effectiveness of our design. We have added this new ablation study to our paper and revised Section 4.4, Table 3, and Appendix C.3 accordingly.
>
> **CVRP (task order 1):**
> |Method|AP|AF|AMF|APl|AG|
> |-|-|-|-|-|-|
> LLR-BC-IB|35.4|23.4|27.2|4.4|31.0
> **LLR-BC**|**4.7**|**0.6**|**0.7**|**4.1**|**23.3**
>
> **TSP (task order 1):**
> |Method|AP|AF|AMF|APl|AG|
> |-|-|-|-|-|-|
> LLR-BC-IB|2.5|1.8|2.0|**1.0**|21.9
> **LLR-BC**|**1.7**|**0.8**|**0.9**|1.3|**21.6**
>
> **Q4: Is there any empirical evidence supporting the claims made in lines 236–240?**
>
>
> Our claims are based on conclusions established in the existing literature. Accordingly, we did not include further empirical experiments dedicated to this aspect in this paper. Apart from the literature cited in our original manuscript (i.e., Wang et al., 2024a; Qi et al., 2025), this conclusion about the characteristics of KLD and RKLD is widely acknowledged in other works, e.g., Section 3.3 of [8] and Section 1 of [9].
>
> [8] Alan Chan, Hugo Silva, Sungsu Lim, Tadashi Kozuno, A. Rupam Mahmood and Martha White. Greedification Operators for Policy Optimization: Investigating Forward and Reverse KL Divergences. Journal of Machine Learning Research, pp. 1-79, 2022.
>
> [9] Taiqiang Wu, Chaofan Tao, Jiahao Wang, Runming Yang, Zhe Zhao, and Ngai Wong. Rethinking Kullback-Leibler Divergence in Knowledge Distillation for Large Language Models. In Proceedings of the 31st International Conference on Computational Linguistics, pp. 5737–5755, 2025.

---

> > ### Comment · Reviewer_iZkt · 2025-11-24
> >
> > Thanks for your rebuttal. I will raise the score accordingly.

---

> > > ### Author Response · Authors · 2025-11-24
> > >
> > > We greatly appreciate the reviewer for reviewing our response, acknowledging the value of our work, and raising the score!

---

### Official Review · Reviewer_9mCf · 2025-10-25

**Soundness:** 2
**Presentation:** 3
**Contribution:** 2
**Rating:** 4
**Confidence:** 4

**Summary:**

This paper presents a lifelong learning framework called LLR-BC for learning to solve Vehicle Routing Problems (VRPs) across different distributions and scales. LLR-BC employs a customized experience replay strategy across sequentially arising tasks to prevent catastrophic forgetting. The approach introduces two key components: a confidence-aware experience weighting (CaEW) module and a decision-seeking behavior consolidation (DsBC) loss based on reverse KL divergence. Results on TSP and CVRP validate the effectiveness of the proposed methods against baselines.

**Strengths:**

* The paper studies an interesting and relatively underexplored area for neural VRP solvers: lifelong learning across diverse problem distributions and scales.
* The presentation and illustrations are clear, making the paper easy to follow.
* The work presents comprehensive experiments covering multiple task orders, diverse distributions and scales, and multiple backbone VRP neural solvers.
* The performance appears promising under the employed metrics.

**Weaknesses:**

* There is some conceptual overlap with existing works, such as Li et al. 2024 and Feng et al. 2025, which address similar lifelong learning setups. It would be useful to directly compare with these existing methods. Also, while the customization of the replay buffer for VRPs is appreciated, the idea of experience replay is a standard technique in lifelong learning literature.
* The paper proposes using the variance of node selection probabilities as a proxy for model confidence. However, this lacks clear empirical evidence or theoretical justification that probability variance truly represents confidence. My concern is that this may be too strong and fails to distinguish between two scenarios: (1) a genuinely low-confidence model with high uncertainty, and (2) a policy that has learned multi-modal decisions where multiple nodes are equally good selections at certain division points with high confidence.
* The design and naming regarding “DsBC” seem confusing to me, as the loss functions in Equations (1) and (2) are relatively straightforward applications of reverse KL divergence. It is unclear what new conceptual insights or contributions DsBC provides here.
* While the experiments are extensive in scope, the absolute performance of the model on each individual task is never reported. In practice, practitioners care not only about whether a lifelong learning model can avoid catastrophic forgetting, but also about the actual solution quality (such as optimality gaps) achieved on each new task.
* The paper lacks a comparison with other state-of-the-art methods that target generalization across distributions without requiring additional training or lifelong learning mechanisms.

**Questions:**

Other than the questions listed in the weakness, I also have the following questions:
* Which epoch of the model training is used to update the buffer experience, and what is the rationale for this choice? How sensitive is performance to this decision?
* How are the hyperparameters for buffer batch size determined?
* How well does variance actually serve as a proxy for confidence? Are there empirical studies or theoretical analyses that validate this design choice against alternative confidence measures?
* How diverse is the experience buffer after running the algorithm? Does the buffer maintain sufficient diversity across tasks, or does it become dominated by certain task types or distributions?

---

> ### Author Response · Authors · 2025-11-21
> **Response to Reviewer 9mCf**
>
> We thank the reviewer for the feedback and would like to address the comments as follows. We have also revised our paper correspondingly, with all revisions highlighted in blue.
>
> **W1: There is some conceptual overlap with existing works, such as Li et al. 2024 and Feng et al. 2025, which address similar lifelong learning setups. It would be useful to directly compare with these existing methods. Also, while the customization of the replay buffer for VRPs is appreciated, the idea of experience replay is a standard technique in lifelong learning literature.**
>
> Although the methods proposed by Li et al., 2024 [1] and Feng et al., 2025 [2] are designed for specific scenarios different from ours and cannot be directly applied to our scenarios, we agree that an experimental comparison with [1] and [2] is helpful. Following the reviewer’s suggestion, **in the revised manuscript we have added comparisons with the methods in [1] and [2], adapted to the scenarios considered in our work**, in which instance generation is uncontrollable. We implemented and adapted these methods because, to the best of our knowledge, no code has been publicly released. Specifically, we introduced instance buffers into their methods and replaced the generation of new instances from previous tasks with reusing buffered instances. We then ran the adapted methods on our scenarios across all six task orders under the same computational budget as LLR-BC. The experimental results, shown below, demonstrate that **LLR-BC generally outperforms their methods in terms of performance on seen tasks (AP), plasticity (APl) for learning new tasks, and generalization to unseen tasks (AG)**. Notably, although the methods of [1] achieve less forgetting (AF, AMF), their overall performance on seen (AP) and unseen (AG) tasks is poor, because a large portion of the training resources (in epochs) on the new task is spent on solving and learning purely from instances of previous tasks, so the new task is not sufficiently learned. This also supports the significance and contribution of our experience replay design.
>
>
>
> **CVRP (mean (std.) over 5 task orders):**
> |Method|AP|AF|AMF|APl|AG|
> |-|-|-|-|-|-|
> Li (inter) [1]|32.0 (6.8)|**0.0** (**0.0**)|**0.0** (**0.0**)|33.6 (7.7)|40.1 (10.5)|
> Li (intra) [1]|34.1 (2.1)|**0.0** (**0.0**)|0.1 (0.1)|39.3 (1.7)|48.2 (4.8)
> Feng [2]|24.6 (2.0)|3.2 (2.0)|4.5 (1.7)|24.2 (0.7)|39.1 (8.6)
> **LLR-BC**|**4.2** (**1.0**)|0.7 (0.5)|0.8 (0.4)|**3.5** (**0.8**)|**26.7** (**4.5**)
>
>
> **TSP (mean (std.) over 5 task orders):**
> |Method|AP|AF|AMF|APl|AG|
> |-|-|-|-|-|-|
> Li (inter) [1]|56.5 (38.3)|0.4 (0.5)| **0.5** (0.5)|61.7 (45.6)|77.6 (59.6)
> Li(intra) [1]|54.2 (7.2)|**0.2** (**0.3**)|**0.5** (**0.4**)|62.8 (12.5)|82.4 (27.7)
> Feng [2]|24.1 (3.2)|1.8 (1.4)|3.8 (1.9)|21.4 (**1.6**)|45.5 (23.2)
> **LLR-BC**|**3.4** (**1.3**)|0.8 (**0.3**)|1.1 (**0.4**)|**2.8** (**1.6**)|**41.1** (**21.0**)
>
> Although experience replay is common in the lifelong learning literature (and [1] and [2] also employ experience replay by generating and solving problem instances of previous tasks), the specific design of the replay mechanism is crucial and can substantially affect performance as shown in above results. We have incorporated the new experimental results into the paper and revised Section 4 and Appendices C and D accordingly.
>
>
> [1] Jingwen Li, Zhiguang Cao, Yaoxin Wu, and Tang Liu. Enhancing the cross-size generalization for solving vehicle routing problems via continual learning, 2024. URL https://openreview.net/forum?id=WdvT2UgsTK.
>
> [2] Shaodi Feng, Zhuoyi Lin, Jianan Zhou, Cong Zhang, Jingwen Li, Kuan-Wen Chen, Senthilnath Jayavelu, and Yew-Soon Ong. Lifelong learner: Discovering versatile neural solvers for vehicle routing problems, 2025. URL https://arxiv.org/abs/2508.11679.

---

> ### Author Response · Authors · 2025-11-21
> **Response to Reviewer 9mCf cnt.**
>
> **W2: The paper proposes using the variance of node selection probabilities as a proxy for model confidence. However, this lacks clear empirical evidence or theoretical justification that probability variance truly represents confidence. My concern is that this may be too strong and fails to distinguish between two scenarios: (1) a genuinely low-confidence model with high uncertainty, and (2) a policy that has learned multi-modal decisions where multiple nodes are equally good selections at certain division points with high confidence.**
>
>
> We thank the reviewer for the thoughtful comment and agree that the two scenarios raised are both possible. However, in our work we use the variance of the action probabilities as a measure of **decision confidence** for a specific state, in order to assess the importance of that state (and its corresponding experience). This is related to, but different from, the overall **model confidence**, which characterizes the model as a whole.
>
> Evaluating state importance via the variance of action probabilities is widely used in the literature [3,4]. We follow this established practice and therefore did not provide additional empirical justification.
>
> In LLR-BC, experiences are buffered at the last training epoch of each task, when we assume the model is already well trained on that task. Under this setting, a state with low **decision confidence** is important in either of the following two cases: (i) high uncertainty indicates that the state is difficult to learn and thus crucial for learning the task [3,4]; (ii) multiple equally good actions, which mainly lead to similarly good solutions, create plateaus in the solution space and make the optimization problem harder to solve [5,6]. Since both types of states are important and should be emphasized, it is not necessary to distinguish them in our current framework. We have added additional discussion about it in the Appendix A.4 in our revised paper. Nevertheless, inspired by the comment, we believe that explicitly differentiating and handling these two cases with more refined mechanisms is a promising direction for future work.
>
>
> [3] Tamilselvan Balasuntharam, Heidar Davoudi and Mehran Ebrahimi. Preferential Proximal Policy Optimization, International Conference on Machine Learning and Applications (ICMLA), pp. 293-300, 2023.
>
> [4] Yitzhak Spielberg and Amos Azaria. The Concept of Criticality in Reinforcement Learning, IEEE International Conference on Tools with Artificial Intelligence (ICTAI), pp. 251-258, 2019.
>
> [5] Sara Tari, Matthieu Basseur and Adrien Goëffon. Sampled Walk and Binary Fitness Landscapes Exploration, Lecture Notes in Computer Science, vol 10764. 2018.
>
> [6] Yiya Diao, Changhe Li, Junchen Wang, Sanyou Zeng and Shengxiang Yang. Bridging the Gap Between Theory and Practice: Fitness Landscape Analysis of Real-World Problems with Nearest-Better Network. Information, 16, 190, 2025.

---

> ### Author Response · Authors · 2025-11-21
> **Response to Reviewer 9mCf cnt.**
>
> **W3: The design and naming regarding “DsBC” seem confusing to me, as the loss functions in Equations (1) and (2) are relatively straightforward applications of reverse KL divergence. It is unclear what new conceptual insights or contributions DsBC provides here.**
>
> Beyond using RKLD as a loss to emphasize the chosen decision (decision-seeking), our Decision-seeking Behavior Consolidation (DsBC) framework also consolidates knowledge from previous tasks via buffered behaviors, i.e., ***behavior consolidation***. Specifically, we leverage the behaviors of past models that are well trained on previous tasks (before forgetting occurs) to guide the consolidation of future models. Rather than relying solely on newly generated instances from old tasks [1,2], we explicitly store the behaviors of historical policies and use them to regularize subsequent training. This design avoids repeatedly generating and solving instances from previous tasks, thereby substantially reducing computational cost and removing the need for controllable instance generation. Moreover, behaviors on previous tasks directly contain the model’s learned knowledge about those tasks, making the consolidation of prior knowledge more straightforward and effective.
>
>
> In this revision, we implemented the methods in [1] and [2] and compared them against LLR-BC, and the experimental results support the performance advantages and contributions of our designs (cf. the response to **W1**). We have also revised Section 4 of the manuscript to clarify these points.
>
>
>
> [7] Alan Chan, Hugo Silva, Sungsu Lim, Tadashi Kozuno, A. Rupam Mahmood and Martha White. Greedification Operators for Policy Optimization: Investigating Forward and Reverse KL Divergences. Journal of Machine Learning Research, pp. 1-79, 2022.
>
> [8] Taiqiang Wu, Chaofan Tao, Jiahao Wang, Runming Yang, Zhe Zhao, and Ngai Wong. Rethinking Kullback-Leibler Divergence in Knowledge Distillation for Large Language Models. In Proceedings of the 31st International Conference on Computational Linguistics, pp. 5737–5755, 2025.
>
>
> **W4: While the experiments are extensive in scope, the absolute performance of the model on each individual task is never reported. In practice, practitioners care not only about whether a lifelong learning model can avoid catastrophic forgetting, but also about the actual solution quality (such as optimality gaps) achieved on each new task.**
>
> We agree that reporting the absolute performance (i.e., solution travel distance) and the optimality gap can further strengthen our experimental study. We have therefore added the absolute performance and optimality gap of each model on each task for all compared methods in Appendix D of the revised manuscript (cf. Tables 6 and 7).

---

> ### Author Response · Authors · 2025-11-21
> **Response to Reviewer 9mCf cnt.**
>
> **W5: The paper lacks a comparison with other state-of-the-art methods that target generalization across distributions without requiring additional training or lifelong learning mechanisms.**
>
>
> In this revision, we further include a comparison with INViT [9], a recent high-performance cross-distribution and cross-scale neural VRP solver that, to the best of our knowledge, represents the current state of the art. Specifically, we directly use the model provided by [9], which is trained with a large computational budget on a uniform-distribution, scale-100 task and demonstrates strong zero-shot generalization to unseen tasks.
>
> The comparison results show that LLR-BC, equipped with lifelong learning, outperforms INViT, which targets cross-distribution generalization without requiring additional training or lifelong learning mechanisms. This further highlights the importance of lifelong learning in our scenario and supports the effectiveness of the proposed LLR-BC. We have added this new experiment to Appendix D.3 of the revised manuscript.
>
>
>
> **Avg. solution distance on test set of each task of CVRP. (Task order 1 for LLR-BC)**
> |Method|E|C|G|U|R|GM
> |-|-|-|-|-|-|-
> INViT|11.1823|15.9606|11.7299|7.0265|5.9726|13.7742
> **LLR-BC**|**10.7198**|**15.1676**|**11.3577**|**6.5640**|**5.6964**|**13.0321**
>
> **Avg. solution distance on test set of each task of TSP. (Task order 1 for LLR-BC)**
> |Method|E|C|G|U|R|GM
> |-|-|-|-|-|-|-
> INViT|5.7489|6.5308|5.6635|3.9311|2.0030|4.5812
> **LLR-BC**|**5.5566**|**6.2116**|**5.4639**|**3.8328**|**1.9755**|**4.1945**
>
> [9] Han Fang, Zhihao Song, Paul Weng, and Yutong Ban. INViT: A generalizable routing problem solver with invariant nested view transformer. In Proceedings of the 41st International Conference on Machine Learning, ICML’24. JMLR, 2024.
>
> **Q1: Which epoch of the model training is used to update the buffer experience, and what is the rationale for this choice? How sensitive is performance to this decision?**
>
> We only buffer newly generated experiences in the last epoch of each task because, by this point, the model has been trained with substantial computational resources and can provide high-quality behaviors and experiences. Such high-quality experiences contain better problem-solving knowledge and better support subsequent consolidation. To examine its effectiveness and the sensitivity to this design, **we further implemented an ablation variant of LLR-BC in which buffering is performed at every epoch, denoted LLR-BC-EE**. The results show that LLR-BC outperforms this ablation variant, further supporting the effectiveness of our design and its importance in LLR-BC. The detailed results are as follows.
>
>
>
>
> **CVRP (task order 1):**
> |Method|AP|AF|AMF|APl|AG|
> |-|-|-|-|-|-|
> LLR-BC-EE|7.8|3.1|3.1|**3.9**|24.1
> **LLR-BC**|**4.7**|**0.6**|**0.7**|4.1|**23.3**
>
> **TSP (task order 1):**
> |Method|AP|AF|AMF|APl|AG|
> |-|-|-|-|-|-|
> LLR-BC-EE|2.6|2.1|2.4|1.6|22.7
> **LLR-BC**|**1.7**|**0.8**|**0.9**|**1.3**|**21.6**
>
> We added the new experimental results to our paper, and revised the Section 4.4, Table 3 and Appendix C.3 correspondingly.
>
>
> **Q2: How are the hyperparameters for buffer batch size determined?**
>
> *“Buffer batch size”* can refer to (i) the buffer size $( |\mathcal{B}| )$, or (ii) the batch size of experiences in the buffer (since we store experiences in units of batches). For (i), we choose the buffer size $( |\mathcal{B}| )$ as large as possible while keeping the additional memory usage acceptable. Ignoring memory constraints, a larger buffer size might generally improve performance because the buffer can contain more diverse experiences. In Section 4.5 of our original paper, we have evaluated different values of $( |\mathcal{B}| )$, and the results indicate that the performance of LLR-BC is not very sensitive to this choice. For (ii), we simply buffer experiences using the same batch size as that used for the task from which the experiences are generated. This is simple and straightforward in our implementation.

---

> ### Author Response · Authors · 2025-11-21
> **Response to Reviewer 9mCf cnt.**
>
> **Q3: How well does variance actually serve as a proxy for confidence? Are there empirical studies or theoretical analyses that validate this design choice against alternative confidence measures?**
>
> As demonstrated in our ablation studies (cf. Section 4.4 and Table 3), weighting experiences by variance works well. To enhance comprehensiveness of our study, **in this revision, we further add comparisons with two additional confidence measures**:
> 1. $(1-h)$, where $h$ is the entropy of the selection probabilities normalized by the maximal possible entropy, similar to the variance normalization in LLR-BC (denoted LLR-BC-Ent).
> 2. the top-2 margin (T2M), i.e., the absolute difference between the largest and second-largest selection probabilities (denoted LLR-BC-T2M). The range of T2M is naturally between 0 and 1.
>
> The three measurements differ in their specific calculations, but they share characteristics that we consider important: they provide higher values when the estimated best action is selected with high probability and reflect the model’s confidence in its estimation of the best action, i.e., decision confidence.
>
> The experimental results are shown below. Although T2M performs slightly worse in terms of forgetting (arguably because it focuses only on the top two actions and thus loses information about the others), the overall performance of these measures is very similar. This suggests that, while variance is a simple and effective choice, LLR-BC is not very sensitive to the specific confidence measure, and other measures with similar properties can also be used. We have added this new ablation study to the paper and revised Section 4.4, Table 3, and Appendix C.3 accordingly.
>
>
>
> **CVRP (task order 1):**
> |Method|AP|AF|AMF|APl|AG|
> |-|-|-|-|-|-|
> LLR-BC-Ent|4.8|**0.5**|**0.7**|4.0|**22.7**
> LLR-BC-T2M|4.8|1.0|1.2|**3.4**|22.8
> **LLR-BC**|**4.7**|0.6|**0.7**|4.1|23.3
>
> **TSP (task order 1):**
> |Method|AP|AF|AMF|APl|AG|
> |-|-|-|-|-|-|
> LLR-BC-Ent|2.1|0.9|1.0|1.6|21.9
> LLR-BC-T2M|**1.7**|0.9|1.1|**1.0**|**21.6**
> **LLR-BC**|**1.7**|**0.8**|**0.9**|1.3|**21.6**
>
> **Q4: How diverse is the experience buffer after running the algorithm? Does the buffer maintain sufficient diversity across tasks, or does it become dominated by certain task types or distributions?**
>
> We record the source task of each experience in the buffer after training on all tasks with one run of LLR-BC under task order 1. With a buffer size of 1000, the numbers of experiences in the buffer from each task are: E: 162, C: 255, G: 141, U: 78, R: 77, GM: 287. The relative frequencies of buffered experiences are highly correlated with the problem scales. This is because an instance with a larger scale generates more experiences (problem-solving steps), and we apply reservoir sampling after each step. As larger-scale tasks are typically more difficult, they may require more experiences to consolidate their knowledge and mitigate catastrophic forgetting. We additionally conduct an experiment with a variant of LLR-BC that rescales the reservoir sampling probability by problem scale, so that the scale no longer affects the source-task ratio in the buffer. After training on all tasks with this rescaled variant (denoted LLR-BC-Res), the numbers of experiences in the buffer from each task are: E: 148, C: 148, G: 144, U: 199, R: 208, GM: 153. Although the buffer becomes more balanced, the performance of this rescaled version (as shown below) is very similar to, and in many cases slightly worse than, the original LLR-BC, further supporting the effectiveness of our design. We have added this new ablation study to the paper and revised Section 4.4, Table 3, and Appendix C.3 accordingly.
>
> **CVRP (task order 1):**
> |Method|AP|AF|AMF|APl|AG|
> |-|-|-|-|-|-|
> LLR-BC-Res|4.9|0.9|0.9|**3.7**|23.9
> **LLR-BC**|**4.7**|**0.6**|**0.7**|4.1|**23.3**
>
> **TSP (task order 1):**
> |Method|AP|AF|AMF|APl|AG|
> |-|-|-|-|-|-|
> LLR-BC-Res|2.0|1.0|1.0|**1.3**|22.3
> **LLR-BC**|**1.7**|**0.8**|**0.9**|**1.3**|**21.6**

---

> ### Author Response · Authors · 2025-11-27
> **Kindly ask for feedback from Reviewer 9mCf**
>
> Dear Reviewer 9mCf,
>
> As the discussion phase is nearing its conclusion, we kindly ask if you could provide feedback on our responses and revisions. If you find that we have satisfactorily addressed your concerns, we would greatly appreciate your consideration of adjusting your rating to reflect the improvements made.
>
> We sincerely thank you for your time and thoughtful review.
>
> Best regards,
>
> Paper 11068 Authors

---

> > ### Comment · Reviewer_9mCf · 2025-11-27
> >
> > I thank the authors for the detailed experiments and the explanation. The response has mostly addressed my concerns regarding W1, W3, W4, and W5.  However, regarding W2, the response has not resolved my concern. I hope to see some clear evidence or analyses to support the claims.

---

> > > ### Author Response · Authors · 2025-11-30
> > > **Response to Reviewer 9mCf**
> > >
> > > We sincerely thank the reviewer for carefully reading our previous response and for acknowledging its value. We also appreciate the additional feedback on W2 and provide further clarification below.
> > >
> > >
> > > ***"The paper proposes using the variance of node selection probabilities as a proxy for model confidence. However, this lacks clear empirical evidence or theoretical justification that probability variance truly represents confidence."***
> > >
> > > In our work, we use the variance of node selection probabilities as a proxy for the **confidence of a specific decision**, i.e., how confident the policy model is in its selected node for a given state. To the best of our knowledge, the term *"model confidence"* is not a standard concept in the DRL literature. We therefore interpret the reviewer’s use of *"model confidence"* as referring to the **confidence of individual decisions** made by a model.
> > >
> > > In the DRL literature, **confidence** is most commonly formalized via the **negative entropy** of the action selection probability distribution (e.g., *"'Confidence' is calculated as negative entropy"* in [1], *"we use the model’s underlying confidence (negative entropy) as a reward"* in [2], *"the negative entropy formulation ensures that lower entropy (higher confidence) receives higher reward"* in [3]). In the following, we therefore take **negative entropy** as the canonical mathematical representation of confidence and show that **variance can represent confidence (negative entropy) very well**, both theoretically and empirically.
> > >
> > > * **Theoretical justification.** Variance and negative entropy share the same qualitative behavior as functions of the selected action probability. For clarity, we consider a binary-action case with actions $a_1$ and $a_2$. Suppose the policy outputs probabilities $p_1 = p$ and $p_2 = 1 - p$. Let $\bar{p} = \frac{p_1 + p_2}{2} = 0.5$ denote the mean probability. Then the variance and confidence (negative entropy) can be written as: $\mathrm{var}(p_1, p_2) = (p - 0.5)^2$, $\mathrm{conf}(p_1, p_2) = p \log p + (1-p)\log(1-p)$.
> > >
> > >   * **Shared monotonic behavior.** The gradients of variance and confidence with respect to $p$ are $\frac{\partial\ \mathrm{var}}{\partial p} = 2p - 1,\quad \frac{\partial\ \mathrm{conf}}{\partial p} = \log\frac{p}{1-p}$. Both gradients are positive when $p > 0.5$ and negative when $p < 0.5$, i.e., both quantities increase as the selected action becomes more likely than all alternatives and decrease when it becomes less likely. Thus, variance and negative entropy exhibit the **same qualitative dependence on the selected action probability**.
> > >   * **Shared minimum.** Both quantities attain their minimum at the uniform distribution $p_1 = p_2 = 0.5$, where the policy is maximally uncertain. This is exactly the situation of **lowest confidence**, and both variance and negative entropy capture it consistently.
> > >
> > >
> > > * **Empirical evidence.** We also observe strong empirical alignment between variance and negative entropy:
> > >   * **High correlation on buffered experiences.** We compute both the variance and the negative entropy for all experiences stored in the buffer after running LLR-BC once under task order 1. The Pearson correlation coefficient between the two measurements is **0.972367** (where 1 indicates perfect positive correlation), showing that, in practice, variance tracks confidence during the lifelong learning process very well.
> > >   * **Similar impact on performance.** As reported in our previous response to W2, we conduct an ablation study comparing the original LLR-BC with LLR-BC-Ent (which uses negative entropy instead of variance). Their performance is very close. This supports the claim that **variance is an effective and a good proxy for confidence**.
> > >
> > > [1] Junhyuk Oh, et al. Discovering state-of-the-art reinforcement learning algorithms. Nature, 2025. https://doi.org/10.1038/s41586-025-09761-x
> > >
> > > [2] Mihir Prabhudesai, et al. Maximizing Confidence Alone Improves Reasoning. arXiv, 2025. https://arxiv.org/abs/2505.22660
> > >
> > > [3] Guobin Shen, et al. Safety Instincts: LLMs Learn to Trust Their Internal Compass for Self-Defense. arXiv, 2025. https://arxiv.org/abs/2510.01088
> > >
> > >
> > > ***"My concern is that this may be too strong and fails to distinguish between two scenarios: (1) a genuinely low-confidence model with high uncertainty, and (2) a policy that has learned multi-modal decisions where multiple nodes are equally good selections at certain division points with high confidence."***
> > >
> > > In our method, the **goal of the confidence measure is to quantify the importance of an experience**, thereby identifying states/behaviors that are more informative or more vulnerable to forgetting in lifelong learning. From this perspective, as discussed in our previous response, our framework does **not require** distinguishing between the two scenarios above. Both are important and should be emphasized during training, which is exactly what LLR-BC does.

---

### Official Review · Reviewer_ckwk · 2025-10-29

**Soundness:** 3
**Presentation:** 3
**Contribution:** 3
**Rating:** 4
**Confidence:** 4

**Summary:**

This paper proposes LLR-BC solvers that address the catastrophic forgetting problem when adapting to sequential tasks with different scales and distributions. It further introduces two key modules: CaEW and DsBC. Moreover, Extensive experiments demonstrate the effectiveness of LLR-BC.

**Strengths:**

1.	The paper introduces two novel modules CaEW and DsBC， with the innovative use of Reverse KL Divergence enabling decision-focused behavior consolidation. It also removes the common “fixed task order” constraint, evaluating on randomized sequential tasks for greater realism and originality.
2.	The model's performance was evaluated across multiple task sequences and VRP scenarios of varying scales and distributions. Furthermore, the paper includes comprehensive ablation experiments and hyperparameter sensitivity analyses, demonstrating the independent effectiveness of the CaEW and DsBC modules as well as the model's robustness to parameter variations.
3.	The presentation is clear and easy to follow.

**Weaknesses:**

1. Limited Innovation: The paper shows limited novelty based on the existing lifelong VRP models. The comparative experiments include only Fine-tuning, EWC, and LiBOG. However, no comparisons are made with more recent lifelong learning approaches (e.g., Li et al., 2024; Feng et al., 2025) or with cross-problem/multi-task frameworks. Including these would better position the proposed method within the current research landscape.
2. Potential Bias in Experience Buffering: Experience buffering is conducted only during the final epoch of each task. A further analysis or ablation study on this design choice would enhance the completeness and reliability of the evaluation.
3. The manuscript contains many typos (e.g., “stuides” → “studies,” “LLB-RC” → “LLR-BC”)

**Questions:**

Clarification on the Logic of the CaEW Mechanism: In Section3.2 , CaEW aims to assign higher weights to “low-confidence” decisions, noting that “lower variance indicates lower confidence.” However, the formula actually assigns higher weights to experiences with low variance (i.e., high confidence). Could you please clarify?

Regarding the limitation of fixed sampling size |ε|: In conclusion that using a fixed |ε| is a limitation when task scales vary significantly. Could you elaborate on why this is problematic? Is it because tasks of different scales require different numbers of experiences for effective consolidation?

---

> ### Author Response · Authors · 2025-11-21
> **Response to Reviewer ckwk**
>
> Thank you for the comments and careful review. We have revised our paper correspondingly, with all revisions highlighted in blue.  Below, please find our point-to-point responses.
>
> **W1: Limited Innovation: The paper shows limited novelty based on the existing lifelong VRP models. The comparative experiments include only Fine-tuning, EWC, and LiBOG. However, no comparisons are made with more recent lifelong learning approaches (e.g., Li et al., 2024; Feng et al., 2025) or with cross-problem/multi-task frameworks. Including these would better position the proposed method within the current research landscape.**
>
> We agree that including these works in our experimental comparison can further improve the comprehensiveness of this study. **In this revision, we therefore adapted the methods of Li et al., 2024 [1] and Feng et al., 2025 [2] to our scenarios, and conducted additional experiments comparing our method against [1,2] as well as two straightforward multi-task frameworks, as detailed below.** In summary, our proposed method outperforms all these approaches in our focused scenarios, further confirming its effectiveness and the contribution of this work.
>
>
> 1. **Existing lifelong VRP solvers (Li et al., 2024 [1] and Feng et al., 2025 [2]).** To the best of our knowledge, no official implementation of [1] and [2] is publicly available, so we implemented them from the original descriptions and adapted them to our setting with uncontrollable instance generation by introducing instance buffers and replacing the generation of new instances from previous tasks with replay of buffered instances. We evaluated them with same experiment setting as LLR-BC, and LLR-BC consistently outperforms them. Although [1] achieves lower forgetting (AF, AMF), its plasticity (APl) is poor because half of the training epochs on each new task are spent exclusively on replayed instances from previous tasks, leading to inferior overall performance on seen (AP) and unseen (AG) tasks. We have incorporated these experiments and updated Section 4 and Appendices C and D accordingly.
>
>     **CVRP (mean (std.) over 5 task orders):**
>     |Method|AP|AF|AMF|APl|AG|
>     |-|-|-|-|-|-|
>     Li (inter) [1]|32.0 (6.8)|**0.0** (**0.0**)|**0.0** (**0.0**)|33.6 (7.7)|40.1 (10.5)|
>     Li (intra) [1]|34.1 (2.1)|**0.0** (**0.0**)|0.1 (0.1)|39.3 (1.7)|48.2 (4.8)
>     Feng [2]|24.6 (2.0)|3.2 (2.0)|4.5 (1.7)|24.2 (0.7)|39.1 (8.6)
>     **LLR-BC**|**4.2** (**1.0**)|0.7 (0.5)|0.8 (0.4)|**3.5** (**0.8**)|**26.7** (**4.5**)
>
>
>     **TSP (mean (std.) over 5 task orders):**
>     |Method|AP|AF|AMF|APl|AG|
>     |-|-|-|-|-|-|
>     Li (inter) [1]|56.5 (38.3)|0.4 (0.5)| **0.5** (0.5)|61.7 (45.6)|77.6 (59.6)
>     Li (intra) [1]|54.2 (7.2)|**0.2** (**0.3**)|**0.5** (**0.4**)|62.8 (12.5)|82.4 (27.7)
>     Feng [2]|24.1 (3.2)|1.8 (1.4)|3.8 (1.9)|21.4 (**1.6**)|45.5 (23.2)
>     **LLR-BC**|**3.4** (**1.3**)|0.8 (**0.3**)|1.1 (0.4)|**2.8** (**1.6**)|**41.1** (**21.0**)
>
>
> 2. **Multi-task solvers.** In our lifelong learning setting, tasks arrive sequentially and future tasks are unknown, so multi-task solvers that train on all tasks simultaneously without forgetting are not directly applicable. To nevertheless provide a multi-task reference, we implemented two straightforward multi-task settings of POMO with batch-level and epoch-level task switching (POMO-MT-BL/EL), trained on all six tasks simultaneously for 1200 epochs, matching the budget of LLR-BC. As reported in Appendix D.3, their overall performance is similar, while LLR-BC achieves better results on the last four CVRP tasks and on all TSP tasks, verifying the effectiveness of LLR-BC. We have added this study to Appendix D.3 of the revised manuscript.
>
>
>     **Test performance on each task of CVRP after training on all tasks. (Task order 1 LLR-BC)**
>
>     |Method|E|C|G|U|R|GM
>     |-|-|-|-|-|-|-
>     POMO-MT-EL|**10.6567**|**15.1016**|11.3768|6.5851|5.7132|13.0378
>     POMO-MT-BL|10.6618|15.1032|11.3594|6.5784|5.7140|13.0448
>     **LLR-BC**|10.7198|15.1676|**11.3577**|**6.5640**|**5.6964**|**13.0321**
>
>     **Test performance on each task of TSP after training on all tasks. (Task order 1 for LLR-BC)**
>
>     |Method|E|C|G|U|R|GM
>     |-|-|-|-|-|-|-
>     POMO-MT-EL|5.6101|6.2478|5.4750|3.8550|1.9765|4.2491
>     POMO-MT-BL|5.5924|6.2359|5.4735|3.8486|1.9762|4.2412
>     **LLR-BC**|**5.5566**|**6.2116**|**5.4639**|**3.8328**|**1.9755**|**4.1945**
>
> 3. **Cross-problem solvers.** Our goal is to develop a model-architecture-agnostic lifelong learning method for tasks with different distributions and scales. Different problems (TSP, CVRP, and VRPTW) they typically require problem-specific model components [3], thus a comparison with cross-problem solvers is beyond the scope of this paper. Nevertheless, we agree that cross-problem solvers are a valuable research direction and plan to explore them in future work.

---

> ### Author Response · Authors · 2025-11-21
> **Response to Reviewer ckwk cnt.**
>
> **W2: Potential Bias in Experience Buffering: Experience buffering is conducted only during the final epoch of each task. A further analysis or ablation study on this design choice would enhance the completeness and reliability of the evaluation.**
>
>
> We only buffer experiences in the last epoch of each task because experiences from a well-trained model (after more epochs of training) contain higher-quality problem-solving knowledge and thus better support subsequent consolidation. We agree that further analysis and ablation studies are helpful. **In this revision, we have added a comparison with an ablation version of LLR-BC in which buffering is conducted at every epoch, denoted LLR-BC-EE.** The results show that LLR-BC outperforms this ablation version, further supporting our design. The detailed results are summarized below. We have incorporated these new experimental results into the paper and revised Section 4.4, Table 3, and Appendix C.3 accordingly.
>
> **CVRP (task order 1):**
> |Method|AP|AF|AMF|APl|AG|
> |-|-|-|-|-|-|
> LLR-BC-EE|7.8|3.1|3.1|**3.9**|24.1
> **LLR-BC**|**4.7**|**0.6**|**0.7**|4.1|**23.3**
>
> **TSP (task order 1):**
> |Method|AP|AF|AMF|APl|AG|
> |-|-|-|-|-|-|
> LLR-BC-EE|2.6|2.1|2.4|1.6|22.7
> **LLR-BC**|**1.7**|**0.8**|**0.9**|**1.3**|**21.6**
>
> **W3: The manuscript contains many typos (e.g., “stuides” → “studies,” “LLB-RC” → “LLR-BC”)**
>
> We sincerely thank the reviewer for the careful reading. We have thoroughly proofread the manuscript and corrected the typos in the revised version.
>
>
>
> **Q1: Clarification on the Logic of the CaEW Mechanism: In Section3.2 , CaEW aims to assign higher weights to “low-confidence” decisions, noting that “lower variance indicates lower confidence.” However, the formula actually assigns higher weights to experiences with low variance (i.e., high confidence). Could you please clarify?**
>
> The reviewer is correct that *'CaEW aims to assign **higher weights** to “**low-confidence**” decisions, noting that “**lower variance** indicates **lower confidence**.”'* This is because, as discussed in Section 3.2, lower-confidence (lower-variance) decisions are more likely to change when training on a new task, which can degrade the solution quality on old tasks. By assigning higher weights to such lower-confidence decisions (and the corresponding states), CaEW encourages the learner to pay more attention to them during consolidation, thereby improving the preservation of performance on previously learned tasks.
>
>
> **Q2: Regarding the limitation of fixed sampling size |ε|: In conclusion that using a fixed |ε| is a limitation when task scales vary significantly. Could you elaborate on why this is problematic? Is it because tasks of different scales require different numbers of experiences for effective consolidation?**
>
> Indeed, larger problem scales induce more decision steps (new experiences) per batch. Here, |ε| denotes the number of old experiences in a batch, so the ratio |ε|/scale reflects the proportion of old versus new experiences used for learning. With a fixed |ε|, this ratio can vary substantially as the scale changes: for very large-scale tasks (e.g., 10K nodes), new experiences dominate and may cause severe forgetting, whereas for very small-scale tasks (e.g., 10 nodes), old experiences dominate and reduce plasticity. Although a fixed |ε| works well in our experiments, we acknowledge that it may become a limitation when task scales differ drastically and plan to refine this design in future work.
>
> [1] Jingwen Li, Zhiguang Cao, Yaoxin Wu, and Tang Liu. Enhancing the cross-size generalization for solving vehicle routing problems via continual learning, 2024. URL https://openreview.net/forum?id=WdvT2UgsTK.
>
> [2] Shaodi Feng, Zhuoyi Lin, Jianan Zhou, Cong Zhang, Jingwen Li, Kuan-Wen Chen, Senthilnath Jayavelu, and Yew-Soon Ong. Lifelong learner: Discovering versatile neural solvers for vehicle routing problems, 2025. URL https://arxiv.org/abs/2508.11679.
>
> [3] Darko Drakulic, Sofia Michel and Jean-Marc Andreoli. GOAL: A Generalist Combinatorial Optimization Agent Learner. In The Thirteenth International Conference on Learning Representations, 2025.

---

> ### Author Response · Authors · 2025-11-27
> **Kindly ask for feedback from Reviewer ckwk**
>
> Dear Reviewer ckwk,
>
> Following your valuable comments, we have carefully revised the manuscript and clarified the points you raised in our responses. With the discussion period approaching its conclusion, we would greatly appreciate it if you could take a moment to re-evaluate the paper in its current form. If you find that your main concerns have been satisfactorily resolved, we would very much appreciate it if you could consider adjusting your score accordingly.
>
> We are thankful for your careful review and for the thoughtful remarks you have provided.
>
> Best regards,
>
> Paper 11068 Authors

---

### Official Review · Reviewer_A4WR · 2025-10-31

**Soundness:** 2
**Presentation:** 3
**Contribution:** 2
**Rating:** 4
**Confidence:** 4

**Summary:**

This paper proposes a lifelong learning framework, LLR-BC, for enhancing the generalization performance of existing neural models for vehicle routing problems (VRPs). It trains multiple tasks (including VRPs with different distributions and sizes) sequentially, and employs experience replay strategy and introduces behavior consolidation to preserve knowledge of prior tasks while learning new ones. Experimental results show the effectiveness of the proposed method.

**Strengths:**

1. This paper employs lifelong learning to solve the generalization issue in VRPs across different distributions and problem sizes.
2. In experiments, this paper introduces interesting evaluation metrics.
3. Experimental results show the effectiveness of the proposed method against baseline methods.

**Weaknesses:**

1. The core idea of this paper, using lifelong learning to address the generalization issue of VRP while alleviating catastrophic forgetting is similar to [1][2]. Although the authors claim that [1][2] focuses on a fixed task order, while the proposed LLR-BC could handle flexible task orders, the core contribution seems to be similar.
2. This paper aims to enhance the generalization ability of neural models for VRPs, while the comparison of the LLR-BC with other neural models that are specialized for generalizability enhancement is not reported.
3. The experience replay strategy is a standard strategy in lifelong learning. And the core idea of behavior consolidation seems to be similar to "inter-task regularization" and "intra-task regularization" in [1], which both learn to preserve knowledge from past ones when training on new tasks. A more clear discrepancy between [1] and this paper should be explained.

[1] Jingwen Li, Zhiguang Cao, Yaoxin Wu, and Tang Liu. Enhancing the cross-size generalization for solving vehicle routing problems via continual learning, 2024. URL https://openreview.net/forum?id=WdvT2UgsTK.

[2] Shaodi Feng, Zhuoyi Lin, Jianan Zhou, Cong Zhang, Jingwen Li, Kuan-Wen Chen, Senthilnath Jayavelu, and Yew-Soon Ong. Lifelong learner: Discovering versatile neural solvers for vehicle routing problems, 2025. URL https://arxiv.org/abs/2508.11679.

**Questions:**

Please refer to the weakness.

---

> ### Author Response · Authors · 2025-11-21
> **Response to Reviewer A4WR**
>
> Thank you for the comments and careful review. We have revised our paper, with all revisions highlighted in blue. Here, we have provided point-to-point responses to all comments.
>
>
> **W1. The core idea of this paper, using lifelong learning to address the generalization issue of VRP while alleviating catastrophic forgetting is similar to [1][2]. Although the authors claim that [1][2] focuses on a fixed task order, while the proposed LLR-BC could handle flexible task orders, the core contribution seems to be similar**
>
> We are grateful to the reviewer for pointing out the similarity between our work and the related literature [1,2]. We summarize the main contributions and novelties that distinguish our work from [1,2]. Then, new results of experimental comparison with [1,2] are reported.
>
> Our contributions and novelties compared to [1,2] lie in two main aspects.
>
>
> **1. Difference in scenarios and assumptions.**
> We consider an online lifelong learning scenario where tasks can differ in both **distribution** and **scale**, arrive in a **random and unknown order**, and must be handled sequentially. Our objective is to improve, at each time step, the solver’s performance on all learned tasks and its generalization to unseen tasks. This is substantially different from [1] and [2], which mainly assume a **known fixed task order** and primarily evaluate the solver **only after learning all tasks**. We have revised Section 1 to clarify these differences in detail:
>
> - **Assumptions on task distribution and scale.** In [1], tasks share the same distribution (node coordinates and demand) and differ only in scale. [2] considers a scenario where tasks share the same distribution and scale but different distance metric, and also the scenario of [1]. In contrast, we do not impose these assumptions: tasks may follow different distributions and have different scales, and the information of each task is unknown until it arrives. Since distribution shifts are common in real-world applications, removing the identical-distribution-and-scale assumption makes our scenario more realistic but also more challenging. Improving neural solvers under distribution shifts is recognized as both difficult and important [3,4,5]. Accordingly, our scenario and contributions are significantly different from those of [1,2]. Consistent with this, the methods in [1,2], which are specifically designed and evaluated only in their scenarios, perform poorly in our scenario, as shown by the new experimental comparisons we added.
> - **Assumptions on task order.**  Both [1] and [2] assume that the task order is fixed and known. In [1], task scales gradually increase, and their method explicitly exploits this order (“This strategy is devised to ensure that the deep model is primarily trained on the current task, i.e., the VRP with a larger size and higher complexity than the previous ones”, Section 4.1 of [1]). [2] targets a specific sequence of changes in the distance metric, and also the scenario of [1]. As task order has a strong impact on lifelong learning performance [6], this is a restrictive assumption. We instead study more general and practical scenarios where the order is random and unknown, and our method is not tailored to any particular order. Our experiments therefore use multiple randomly generated task orders for a comprehensive evaluation, and verify the superior performance of our method.
> - **Assumptions on control over problem instance generation.** Both [1] and [2] assume that the learner can actively generate new training instances from previous tasks. In our scenario, the lifelong learner has no control over instance generation. Consequently, **their methods that rely on generating new problem instances for selected past tasks cannot be directly applied in our scenarios**.
> - **Assessment protocol.** To match our objective, we evaluate the solver’s performance on each task both before and after training on it. By contrast, [1] and [2] mainly test and report performance only after training on all tasks, i.e., at the end of the lifelong learning process, without intermediate evaluation. Taken together with the above assumptions, [1] and [2] are closer to curriculum learning methods, which aim to obtain a strong final model under a specific training order (e.g., from easy to hard). In contrast, our work evaluates performance both during and at the end of learning, proposes a comprehensive set of metrics, and reports detailed results and analysis over the entire lifelong learning process.
>
>
> The aforementioned differences in scenarios are summarized below:
>
> | Work  | Task difference | Task order | Control over instance generation
> | ---- | - | ------- |------- |
> | [1] | scale  | fixed, scale gradually increases | Have control
> | [2]   | scale or distance metric  | fixed, scale gradually increases / Euclidean→Manhattan→Chebyshev | Have control
> | ours | distribution and scale  | random and unknown | No control

---

> ### Author Response · Authors · 2025-11-21
> **Response to Reviewer A4WR cnt.**
>
> **2. Difference in method design.** Owing to the different scenarios, our method departs substantially from [1] and [2] in the following aspects, which explains its superior performance in our scenario (as shown in the following additional experiments). These differences are further clarified in Section 3 of the revised paper.
> - **Formulation of experience.** We treat each step (a ⟨state, behavior⟩ pair) in solving an instance as one experience, whereas [1,2] treat the entire instance as one experience. This step-based formulation provides finer granularity and is more effective under the same computational budget. We conducted an additional ablation study (Section 4.4, Table 3) comparing LLR-BC with an instance-based variant (LLR-BC-IB), and the results confirm the superior effectiveness of our step-based design.
> - **Additional cost for generating experience.** Our method reuses previously buffered experiences for consolidation instead of regenerating new ones as in [1,2]. This reduces computational cost and also makes our approach applicable to scenarios where instance generation is uncontrollable, in which **the methods of [1,2] cannot be directly applied**.
> - **How consolidation is performed.** To preserve previously learned knowledge when learning a new task, [2] **regularizes the parameters** of the current model to stay close to those of the previous model, in a way similar to EWC. As shown in experiments of our original paper (cf. Table 1), parameter-level regularization (EWC, LiBOG) leads to inferior performance. The method in [1] encourages the current and previous models to behave similarly **on the new task**. In contrast, our method encourages the current and previous models to exhibit similar behavior **on the previous tasks**. Behaviors on previous tasks encode more direct knowledge about those tasks and therefore provide more effective signals for consolidation. Our method thus achieves better performance than [1] and [2] in the following reported experiments.
> - **Additional design components.** In addition, we introduce a new confidence-based weighting scheme and an new RKLD-based decision-seeking consolidation objective, as described in the original manuscript (Sections 3.2 and 3.3). The new components further improve the performance of our method (cf. Section 4.4 and Table 3 of the original paper).
>
>
> | Work  | Formulation of experience | Replay buffer | Source of guiding information for consolidation
> | ---- | - | ------- |-------  |
> | [1] | whole instance  | no, will sample new instances of previous task  | behavior of previous model on new task
> | [2]   | whole instance  | no, will sample new instances of previous task  | previous model's parameter values
> | Ours | pair <state, behavior>  | yes | behavior of previous model on previous task
> | Our motivation/advantage |leverage information contained in the solver's behavior | applicable when instance generation is uncontrollable and reduce computational cost| high quality knowledge about previous task, more exploration in parameter space  |
>
>
>
>
> [1] Jingwen Li, Zhiguang Cao, Yaoxin Wu, and Tang Liu. Enhancing the cross-size generalization for solving vehicle routing problems via continual learning, 2024. URL https://openreview.net/forum?id=WdvT2UgsTK.
>
> [2] Shaodi Feng, Zhuoyi Lin, Jianan Zhou, Cong Zhang, Jingwen Li, Kuan-Wen Chen, Senthilnath Jayavelu, and Yew-Soon Ong. Lifelong learner: Discovering versatile neural solvers for vehicle routing problems, 2025. URL https://arxiv.org/abs/2508.11679.
>
> [3] Yuan Jiang, Yaoxin Wu, Zhiguang Cao, &andJie Zhang. Learning to Solve Routing Problems via Distributionally Robust Optimization. Proceedings of the AAAI Conference on Artificial Intelligence, 36(9), 9786-9794, 2022. https://doi.org/10.1609/aaai.v36i9.21214
>
> [4] Jianan Zhou, Yaoxin Wu, Wen Song, Zhiguang Cao, and Jie Zhang. Towards omni-generalizable neural methods for vehicle routing problems. Proceedings of the 40th International Conference on Machine Learning, volume 202 of Proceedings of Machine Learning Research, pp. 42769–42789. PMLR, 23–29 Jul 2023.
>
> [5] Han Fang, Zhihao Song, Paul Weng, and Yutong Ban. INViT: A generalizable routing problem solver with invariant nested view transformer. In Proceedings of the 41st International Conference on Machine Learning, ICML’24. JMLR, 2024.
>
> [6] Samuel J. Bell, and Neil D. Lawrence. The Effect of Task Ordering in Continual Learning，2022. URL https://arxiv.org/abs/2205.13323

---

> ### Author Response · Authors · 2025-11-21
> **Response to Reviewer A4WR cnt.**
>
> ### **New experimental results (compared to [1,2])**
>
> We adapted the methods in [1] and [2] to our scenarios by introducing instance buffers, thereby removing their reliance on controllable generation of instances from previous tasks, and then conducted additional experiments. The results (all five metrics are minimization metrics, i.e., smaller values indicate better performance, cf. Section 4) show that our method significantly outperforms the methods of [1] and [2] in our scenarios (Tables 1, 2 and Figures 3, 4 in the revised paper), further supporting its effectiveness. Notably, although [1] achieves the best (smallest) AF and AMF scores (less forgetting), its plasticity (APl) for learning new tasks is poor (large), because half of the training budget (in epochs) on each new task is spent exclusively on instances sampled from previous tasks. As a result, its overall performance on seen tasks (AP) and unseen tasks (AG) is inferior.
>
>
> **CVRP (mean (std.) over 5 task orders):**
> |Method|AP|AF|AMF|APl|AG|
> |-|-|-|-|-|-|
> Li (inter) [1]|32.0 (6.8)|**0.0** (**0.0**)|**0.0** (**0.0**)|33.6 (7.7)|40.1 (10.5)|
> Li (intra) [1]|34.1 (2.1)|**0.0** (**0.0**)|0.1 (0.1)|39.3 (1.7)|48.2 (4.8)
> Feng [2]|24.6 (2.0)|3.2 (2.0)|4.5 (1.7)|24.2 (0.7)|39.1 (8.6)
> **LLR-BC**|**4.2** (**1.0**)|0.7 (0.5)|0.8 (0.4)|**3.5** (**0.8**)|**26.7** (**4.5**)
>
> **TSP (mean (std.) over 5 task orders):**
> |Method|AP|AF|AMF|APl|AG|
> |-|-|-|-|-|-|
> Li (inter) [1]|56.5 (38.3)|0.4 (0.5)| **0.5** (0.5)|61.7 (45.6)|77.6 (59.6)
> Li (intra) [1]|54.2 (7.2)|**0.2** (**0.3**)|**0.5** (**0.4**)|62.8 (12.5)|82.4 (27.7)
> Feng [2]|24.1 (3.2)|1.8 (1.4)|3.8 (1.9)|21.4 (**1.6**)|45.5 (23.2)
> **LLR-BC**|**3.4** (**1.3**)|0.8 (**0.3**)|1.1 (0.4)|**2.8** (**1.6**)|**41.1** (**21.0**)

---

> ### Author Response · Authors · 2025-11-21
> **Response to Reviewer A4WR cnt.**
>
> **W2. This paper aims to enhance the generalization ability of neural models for VRPs, while the comparison of the LLR-BC with other neural models that are specialized for generalizability enhancement is not reported.**
>
> Thank you for the comment. Our original submission already included comparisons with neural solvers specialized in cross-distribution and cross-scale generalization, namely Omni [4] and INViT [5] (cf. Section 4.6, Table 4, and Appendix D.6). To the best of our knowledge (up to the ICLR 2026 submission deadline), no existing method directly targets our scenario, i.e., lifelong learning on VRP tasks differing in both scale and distribution. We therefore selected Omni and INViT, two recent and closely related neural solvers originally designed to enhance generalization without lifelong learning, and adapted them to our scenario for comparison.
>
> We additionally involve a baseline where INViT [5] is trained in its original non-lifelong setting (c.f Appendix D.3), to provide a more comprehensive comparison. Experimental results are as follows.
>
> **Avg. solution distance on test set of each task of CVRP. (Task order 1 for LLR-BC)**
> |Method|E|C|G|U|R|GM
> |-|-|-|-|-|-|-
> INViT|11.1823|15.9606|11.7299|7.0265|5.9726|13.7742
> **LLR-BC**|**10.7198**|**15.1676**|**11.3577**|**6.5640**|**5.6964**|**13.0321**
>
> **Avg. solution distance on test set of each task of TSP. (Task order 1 for LLR-BC)**
> |Method|E|C|G|U|R|GM
> |-|-|-|-|-|-|-
> INViT|5.7489|6.5308|5.6635|3.9311|2.0030|4.5812
> **LLR-BC**|**5.5566**|**6.2116**|**5.4639**|**3.8328**|**1.9755**|**4.1945**
>
> Taking both the original and new results together (cf. Section 4.6, Tables 4, 6, 7, and Appendix D.3 and D.6), our LLR-BC consistently outperforms existing lifelong-learning neural VRP solvers [1,2], widely used strategies such as fine-tuning and EWC, as well as INViT without lifelong mechanisms. These results further support both the importance of lifelong learning in this scenario and the effectiveness of the proposed LLR-BC.
>
> Taking both the original and new results together (cf. Section 4.6, Tables 4, 6, 7, and Appendix D.3 and D.6), our LLR-BC consistently outperforms existing lifelong-learning neural VRP solvers [1,2], widely used strategies such as fine-tuning and EWC, as well as INViT without lifelong mechanisms. These results further support both the importance of lifelong learning in this scenario and the effectiveness of the proposed LLR-BC.
>
>
> **W3: The experience replay strategy is a standard strategy in lifelong learning. And the core idea of behavior consolidation seems to be similar to "inter-task regularization" and "intra-task regularization" in [1], which both learn to preserve knowledge from past ones when training on new tasks. A more clear discrepancy between [1] and this paper should be explained.**
>
> The detailed discrepancies between [1] and our work have been thoroughly discussed in our response to **W1**, and we have revised the paper accordingly for further clarification. As demonstrated by the new experimental results, our design is not only novel but also effective.

---

> ### Author Response · Authors · 2025-11-27
> **Kindly ask for feedback from Reviewer A4WR**
>
> Dear Reviewer A4WR,
>
> As the discussion is nearing its end, we kindly invite you to take a fresh look at the revised manuscript and our responses. If, upon reconsideration, you feel that the paper has improved in line with your expectations and that your earlier concerns have been adequately addressed, we would be thankful if you could consider adjusting your score accordingly.
>
> We sincerely thank you for your time and thoughtful review.
>
> Best regards,
>
> Paper 11068 Authors

---

### Author Response · Authors · 2025-12-02
**Response to AC**

Dear AC,

Thank you for coordinating the review process at this stage, and we are also grateful to the previous AC for the efforts in overseeing the earlier phases of the review. Below we briefly summarize the main concerns, our responses, and the corresponding revisions. All changes are highlighted in blue in the revised manuscript.

**We also note that both reviewers who have responded to our rebuttal (iZkt and 9mCf) provided very positive comments on our response and the revised paper.** Reviewer iZkt explicitly stated that all concerns have been addressed and **has already increased the score**. Reviewer 9mCf stated that (at that time, we believe the score editing function was already closed) all concerns except W2 have been addressed, and requested further analysis of W2. We have responded and provided comprehensive theoretical justification and empirical evidence to specifically address W2.

---

### 1. Novelty and difference w.r.t. existing lifelong learning solvers [1,2], and experimental comparison with generalizable VRP solvers



Reviewers (A4WR, ckwk, 9mCf, iZkt) asked us to clarify our novelty relative to [1] and [2], and to compare against their methods as well as recent generalizable neural VRP solvers. (This is the major concern of all reviewers. Both reviewers who have responded to our rebuttal (iZkt and 9mCf) explicitly state that this concern has been addressed by our further clarification and additional experimental studies.)

- We have clearly distinguished our **scenario** from [1,2]: in our work, tasks differ in both **distribution and scale**, arrive in a **random and unknown order**, instance generation is **uncontrollable**, and performance is evaluated **throughout** the lifelong process. In contrast, [1,2] assume a fixed known task order, shared distributions/scales and controllable instance generation, and primarily evaluate only at the end of training.
- We have clarified the **methodological novelty**: step-based experiences (⟨state, behavior⟩), buffer-based behavior consolidation on **previous tasks**, reuse of buffered experiences instead of generating new ones, and a confidence-based weighting scheme combined with an RKLD-based decision-seeking objective.
- We have implemented and adapted the methods of [1] and [2] (no official code is available) to our scenarios via instance buffers. **We have added comprehensive experimental comparisons with their methods. Across CVRP and TSP and all task orders, LLR-BC achieves substantially better overall performance.**
- Beyond our original comparison with Omni (a recent meta-learning solver) and INViT (a recent and strong generalizable solver) in the lifelong setting, **we further added comparisons with INViT without lifelong mechanisms, as well as with two multi-task POMO variants (POMO-MT-EL/BL). LLR-BC matches or outperforms these baselines, highlighting both the benefit of lifelong learning and the effectiveness of LLR-BC.**

[1] Jingwen Li, et al. Enhancing the cross-size generalization for solving vehicle routing problems via continual learning, 2024. https://openreview.net/forum?id=WdvT2UgsTK.

[2] Shaodi Feng, et al. Lifelong learner: Discovering versatile neural solvers for vehicle routing problems, 2025. https://arxiv.org/abs/2508.11679.


### 2. Experience buffer design

Reviewers (ckwk, 9mCf, iZkt) asked for motivation clarification and analysis of our buffer design, in particular: buffering experiences only in the final epoch of each task, and using step-level rather than instance-based experiences. (Again, both reviewers who have responded explicitly state that this concern has been addressed.)

We revised the paper to emphasize the design motivations (i.e., getting high-quality behaviors and increasing instance diversity of the buffer) and added the following ablation studies for further investigation:
  - **Buffer timing:** LLR-BC-EE (buffer at every epoch) performs noticeably worse, confirming the effectiveness of our design.
  - **State-based vs instance-based buffer:** LLR-BC-IB (instance-based experiences) performs noticeably worse, supporting our step-based design.
  - **Buffer diversity:** We measured the task diversity in the buffer and introduced a rescaled variant (LLR-BC-Res) with more balanced task frequencies. Its worse performance supports our original design.

### 3. Writing and reporting improvements

Finally, we:
- Proofread, corrected typos and improved explanations of reservoir sampling, KLD vs RKLD, and notation of figures.
- Added tables reporting **absolute solution distances and optimality gaps** on each task for all methods (Appendix D), addressing concerns about practical performance.

---

### Summary

Overall, we believe our further clarification and additional experimental studies successfully addressed all the concerns of reviewers. These revisions substantially strengthen the technical clarity, empirical support, and presentation of our work.

Sincerely,

Paper 11068 Authors

---

### Meta-Review · Area_Chair_LfEU · 2026-01-06

**Summary:**

Reviewers' concerns are mainly around the following points:

W1. The core idea of lifelong learning for neural solver is not new, and the novelty is limited

W2. Insufficient comparison to other generalizability enhancement methods

W3. The experience replay strategy is a standard strategy in lifelong learning

W4. Experience buffering may introduce potential bias

W5. Using node selection probabilities as a proxy for model confidence lacks justification

W6. The absolute performance of the model on each individual task is not reported

**Reviewer Concerns:**

All the above concerns have been addressed by the authors' rebuttal. I believe there are no more outstanding concern.

**Reviewer Scores:**

Reviewer iZkt raised his/her score to 6. For other reviewers, I think the raised concerns can be addressed by authors' explanation, especially the clarification on the novelty and differences w.r.t existing works, and the additional comparison to more recent baselines. I believe all reviewers will raise to 6 eventually.

---

### Decision · Program_Chairs · 2026-01-26

Accept (Poster)